SciPost Physics

Submission

# Entanglement spreading after local fermionic excitations in the XXZ chain

Matthias Gruber, Viktor Eisler

Institute of Theoretical and Computational Physics, Graz University of Technology,
NAWI Graz, Petersgasse 16, 8010 Graz, Austria

October 19, 2020

## 1 Abstract

We study the spreading of entanglement produced by the time evolution of a local fermionic excitation created above the ground state of the XXZ chain. The resulting entropy profiles are investigated via density-matrix renormalization group calculations, and compared to a quasiparticle ansatz. In particular, we assume that the entanglement is dominantly carried by spinon excitations traveling at different velocities, and the entropy profile is reproduced by a probabilistic expression involving the density fraction of the spinons reaching the subsystem. The ansatz works well in the gapless phase for moderate values of the XXZ anisotropy, eventually deteriorating as other types of quasiparticle excitations gain spectral weight. Furthermore, if the initial state is excited by a local Majorana fermion, we observe a nontrivial rescaling of the entropy profiles. This effect is further investigated in a conformal field theory framework, carrying out calculations for the Luttinger liquid theory. Finally, we also consider excitations creating an antiferromagnetic domain wall in the gapped phase of the chain, and find again a modified quasiparticle ansatz with a multiplicative factor.

---

# 1   Introduction

The non-equilibrium dynamics of integrable models has developed into a vast field of research [1]. Among the numerous aspects, the understanding of local relaxation and equilibration in closed quantum systems has become a central topic of investigation [2,3]. In this respect, integrable systems show a rather peculiar behaviour, as the dynamics is characterized by the existence of stable quasiparticle excitations. This is intimately related to the extensive number of nontrivial conservation laws, which nevertheless allow for a local relaxation in a generalized sense [4].

Starting from the early studies of this topic, it was identified that the spreading of entanglement must play a key role in our understanding of integrable dynamics. Ground states of homogeneous, local Hamiltonians have a low amount of entanglement, typically satisfying an area law [5]. However, considering the time evolution with respect to a different Hamiltonian as in the context of a global quantum quench [6], the rapid *linear* growth of entanglement was attributed to the ballistic propagation of entangled quasiparticle pairs [7]. These quasiparticles transmit entanglement over large distances, contributing to the buildup of an extensive entropy within any given subsystem, which signals the onset of some local thermalization. Specifically, in one-dimensional integrable chains it has been verified that the entanglement entropy accumulated in a subsystem actually plays the role of the thermal entropy as described by the generalized Gibbs ensemble [8–10].

The global quench is the simplest representative of an initial state that has an extensive amount of energy above the ground state of the Hamiltonian governing the dynamics, thus acting as a reservoir of quasiparticle excitations. The interpretation, however, becomes more complicated if the initial state lies in the low-energy regime. A particular example is the local quench, where the final Hamiltonian is disturbed only locally with respect to the initial one, such as joining two initially separated quantum chains. At criticality, the entanglement spreading can be captured via conformal field theory (CFT) [11–13], predicting a slow *logarithmic* growth of the entropy, which was indeed observed in free-fermion chains [14]. However, despite signatures of the underlying quasiparticle dynamics, such as a light-cone spreading with the maximal group velocity, it is unclear how the individual quasiparticles contribute to the entropy.

Yet another situation that has been studied intensively within CFT is the so-called local operator excitation [15–17]. Here the low-energy initial state is excited from the vacuum of the CFT by the insertion of a local primary operator, while the Hamiltonian is left untouched. The disturbance has then a linear propagation, increasing the entanglement of a segment only while passing through it, with a *constant* excess entropy determined by the quantum dimension of the local primary [15–17]. The calculations have been extended in various directions, considering fermionic [18] or descendant fields [19, 20], multiple excitations [21], as well as the effects of finite temperatures [22] or boundaries [23].

Despite this increased attention, there have been much less studies on entanglement spreading after local excitations in integrable quantum chains. The CFT predictions have been tested on the critical transverse Ising [24] and XX chains [25], for various local operators that are lattice analogs of primary or descendant fields. On the other hand, entanglement spreading has also been considered in the non-critical ordered phase of the

Ising [26] and XY chains [27,28], starting from a domain-wall initial state excited by a local Majorana operator. Remarkably, the emerging profile of the excess entropy was shown to be captured by a simple probabilistic quasiparticle ansatz [28]. Indeed, taking into account the dispersive spreading of quasiparticles, only a certain fraction of the initially localized excitation will cross the subsystem boundary located at a certain distance. Interpreting this quasiparticle fraction as the probability of finding the excitation within the subsystem, the excess entropy is simply given by a binary expression [28].

Here we aim to extend the quasiparticle description of entanglement spreading to local fermionic excitations in the XXZ chain. Being a Bethe ansatz integrable interacting model [29, 30], its quasiparticle content is much more complex than in the free-fermion systems considered so far. Nevertheless, since our local excitations probe the low-energy physics, it seems reasonable that the dominant weight is carried by low-lying spinon excitations, which we shall assume to build our quasiparticle ansatz. Compared against the profiles of the excess entropy, as obtained from density-matrix renormalization group (DMRG) calculations [31–33], we observe a good agreement after a local fermion creation for moderate values of the interaction. For larger interactions in the gapless phase, one finds deviations that can be attributed to different types of quasiparticles with higher energy.

We also study the profiles after a local Majorana excitation, which seem to be given by a simple rescaling of the spinon ansatz. This result is supplemented by CFT calculations carried out for the Luttinger liquid theory, which describes the low-energy physics of the XXZ chain. We find that, due to the left-right mixing of the chiral bosonic modes, the asymptotic excess entropy is doubled for the Majorana excitation, although with a very slow convergence towards this value. Finally, in the gapped phase of the chain we study the excess entropy profile after a local Majorana operator that excites an antiferromagnetic domain wall. Here our numerical results suggest that the spinon ansatz is multiplied by a nontrivial factor, related to the ground-state entropy.

The rest of the manuscript is structured as follows. In section 2 we introduce the XXZ chain and discuss its low-lying excitations. Section 3 is devoted to the study of entanglement spreading after local excitations in the gapless phase: we first introduce a quasiparticle ansatz for the excess entanglement, followed by our numerical studies of a fermion creation as well as a Majorana excitation. Our results for the gapless regime are complemented by a calculation of the Rényi entropy within a CFT framework in section 4. Finally, in section 5 we consider entanglement and magnetization profiles after a domain-wall excitation in the gapped regime. Our closing remarks are given in section 6, followed by an appendix containing the details of the CFT calculations.

## 2    XXZ chain and low-energy excitations

We consider an XXZ chain of length $L$ with open boundary conditions that is given by the Hamiltonian

$$H = J \sum_{j=-L/2+1}^{L/2-1} \left( S_j^x S_{j+1}^x + S_j^y S_{j+1}^y + \Delta S_j^z S_{j+1}^z \right) , \tag{1}$$

where $S_j^\alpha = \sigma_j^\alpha/2$ are spin-1/2 operators acting on site $j$, and $\Delta$ is the anisotropy. The energy scale is set by the coupling $J$ which we fix at $J = 1$. The XXZ Hamiltonian (1) conserves the total magnetization $S^z$ in $z$-direction and we will be interested in its ground state in the zero-magnetization sector $S^z = 0$. Equivalently, the XXZ spin chain can be

122  rewritten in terms of spinless fermions by performing a Jordan-Wigner transformation,
123  which brings (1) into the form

$$H = \sum_{j=-L/2+1}^{L/2-1} \left[ \frac{1}{2}(c_j^\dagger c_{j+1} + c_{j+1}^\dagger c_j) + \Delta \left( c_j^\dagger c_j - \frac{1}{2} \right) \left( c_{j+1}^\dagger c_{j+1} - \frac{1}{2} \right) \right], \tag{2}$$

124  where $c_j^\dagger$ $(c_j)$ are fermionic creation (annihilation) operators, satisfying anticommutation
125  relations $\{c_i, c_j^\dagger\} = \delta_{ij}$. One then has a half-filled fermionic hopping chain with nearest-
126  neighbour interactions of strength $\Delta$. For $|\Delta| \leq 1$ the system is in a critical phase with
127  gapless excitations above the ground state, whereas a gap opens for $|\Delta| > 1$. The case
128  $\Delta = 1$ corresponds to the isotropic Heisenberg antiferromagnet.

129     In the following we give a short and non-technical introduction to the construction
130  of the ground state and low-lying excited states of the XXZ chain. To keep the discus-
131  sion simple, we shall rather consider a periodic chain, and focus on the behaviour in the
132  thermodynamic limit $L \to \infty$. The exact eigenstates of the XXZ chain can be found
133  from Bethe ansatz [29, 30]. These are constructed as a superposition of plane waves, the
134  so-called magnons, labeled by their rapidities $\lambda_i$ which provide a convenient parametriza-
135  tion of the quasimomenta. The allowed values of the rapidities follow from the Bethe
136  equations, with real solutions corresponding to spin-wave like states. Complex solutions
137  organize themselves into strings and correspond to bound states.

138     For $|\Delta| < 1$ the half-filled ground state is obtained by occupying all the allowed
139  vacancies of the $L/2$ real rapidities, thus forming a tightly packed Fermi sea. Low-energy
140  excitations in the $S^z = 1$ sector are called spinons and are created by removing a rapidity.
141  This creates two holes in the Fermi sea, with all the remaining rapidities moving slightly
142  with respect to their ground-state values, and the energy difference can be calculated from
143  this back-flow effect. In the thermodynamic limit, the result can be found analytically and
144  written directly in terms of the quasimomenta $q_1$ and $q_2$ of the two spinons as [29]

$$\Delta E = \varepsilon_s(q_1) + \varepsilon_s(q_2), \tag{3}$$

145  where the spinon dispersion relation in the gapless regime with $\Delta = \cos(\gamma)$ is given by

$$\varepsilon_s(q) = \frac{\pi}{2} \frac{\sin(\gamma)}{\gamma} \sin(q). \tag{4}$$

146  Note that spinons are always excited in pairs, with the individual momenta confined to
147  $0 \leq q_{1,2} \leq \pi$. The total momentum is then given by $q_1 + q_2$, and due to the additivity
148  of (3) one actually has a band of excitation energies. In particular, the lower edge of the
149  two-spinon band is obtained by setting $q_2 = 0$ or $q_2 = \pi$, and thus simply corresponds to
150  shifting the dispersion in (4) for $q > \pi$. The group velocity of the spinons can be directly
151  obtained from the derivative of the dispersion

$$v_s(q) = \frac{\mathrm{d}\varepsilon_s(q)}{\mathrm{d}q} = \frac{\pi}{2} \frac{\sin(\gamma)}{\gamma} \cos(q). \tag{5}$$

152     Further low-energy excitations with $S^z = 1$ can be created by removing a single rapidity
153  from the real axis and placing it onto the $\mathrm{Im}\,\lambda = \pi$ axis. The energy of this particle-hole
154  excitation can be obtained, similarly to the spinon case, from the back-flow equations of
155  the rapidities and yields the dispersion [29]

$$\varepsilon_{ph}(q) = \pi \frac{\sin(\gamma)}{\gamma} \left| \sin\left(\frac{q}{2}\right) \right| \sqrt{1 + \cot^2\left(\frac{\pi}{2}\left(\frac{\pi}{\gamma} - 1\right)\right) \sin^2\left(\frac{q}{2}\right)}. \tag{6}$$

However, in contrast to spinons, particle-hole excitations are not composite objects and their momentum range is thus $0 \leq q < 2\pi$. Note that these spin-wave like excitations are only physical for $-1 < \Delta < 0$, i.e. in case of attractive interactions. For low momenta $q \to 0$, the dispersion relation Eq. (6) approaches the one for spinons in Eq. (4). The group velocities of particle-hole excitations are obtained by taking the derivative of $\varepsilon_{ph}(q)$. Interestingly, it was found that the maximum particle-hole velocity can exceed the maximum spinon velocity only if the anisotropy satisfies $\Delta < \Delta^* \approx -0.3$, which was demonstrated in a particular quench protocol [34].

Finally, we consider the gapped phase where we focus exclusively on the antiferromagnetic regime $\Delta > 1$, with the standard parametrization $\Delta = \cosh \phi$. For even $L$ the ground state has $S^z = 0$ and is again given by $L/2$ magnons with real rapidities. However, the allowed number of vacancies is now $L/2 + 1$, which allows to construct a slightly shifted Fermi sea. In the Ising limit $\Delta \to \infty$, this yields an exact twofold degenerate ground state, given by the linear combinations of the two Néel states

$$|\psi_\pm\rangle = \frac{|\uparrow\downarrow\uparrow\downarrow\ldots\rangle \pm |\downarrow\uparrow\downarrow\uparrow\ldots\rangle}{\sqrt{2}}. \tag{7}$$

For finite $\Delta$, the two states $|\psi_\pm\rangle$ constructed this way are only quasi-degenerate, with an energy difference decaying exponentially in the system size $L$. Considering the thermodynamic limit one can write

$$|\psi_\pm\rangle = \frac{|\psi_\uparrow\rangle \pm |\psi_\downarrow\rangle}{\sqrt{2}}, \tag{8}$$

where $|\psi_\uparrow\rangle$ and $|\psi_\downarrow\rangle$ correspond to ground states with spontaneously broken symmetry, displaying antiferromagnetic ordering. In fact, the bulk expectation value of the staggered magnetization can be calculated analytically as [35, 36]

$$\langle\psi_\uparrow| \sigma_j^z |\psi_\uparrow\rangle = - \langle\psi_\downarrow| \sigma_j^z |\psi_\downarrow\rangle = (-1)^j \prod_{n=1}^{\infty} \tanh^2(n\phi) . \tag{9}$$

The low-lying excitations in the gapped phase are given again by spinons, by creating two holes in the Fermi sea. The excitation energy is still given by Eq. (3), with the dispersion in the gapped phase obtained as [29]

$$\varepsilon_s(q) = \frac{\sinh(\phi)}{\pi} K(u) \sqrt{1 - u^2 \cos^2(q)} , \tag{10}$$

where the complete elliptic integral of the first kind reads

$$K(u) = \int_0^{\pi/2} \frac{\mathrm{d}p}{\sqrt{1 - u^2 \sin^2(p)}} \tag{11}$$

and the elliptic modulus $u$ satisfies

$$\phi = \pi \frac{K(\sqrt{1 - u^2})}{K(u)} . \tag{12}$$

The spinon velocity is obtained from the derivative of (10) and reads

$$v_s(q) = \frac{\sinh(\phi)}{\pi} K(u) \frac{u^2 \sin(q) \cos(q)}{\sqrt{1 - u^2 \cos^2(q)}} . \tag{13}$$

## 3 Entanglement dynamics in the gapless phase

The goal of this section is to study the entanglement dynamics after a particular class of excitations. Namely, we first initialize the chain in its gapless ground state $|\psi_0\rangle$, which is then excited by an operator that is strictly local in terms of the creation/annihilation operators $c_j^\dagger$ and $c_j$ appearing in the fermionic representation (2) of the XXZ chain. The system is then let evolve freely and we are interested in the emerging entanglement pattern in the time-evolved state $|\psi(t)\rangle$. For a bipartition into a subsystem $A$ and the rest of the chain $B$, this is characterized by the von Neumann entropy

$$S(t) = -\text{Tr}\left[\rho_A(t) \ln \rho_A(t)\right], \tag{14}$$

with the reduced density matrix $\rho_A(t) = \text{Tr}_B \rho(t)$ and $\rho(t) = |\psi(t)\rangle \langle\psi(t)|$. In particular, we consider the bipartition $A = [-L/2 + 1, r]$ and $B = [r + 1, L/2]$ and study the entropy profiles

$$\Delta S = S(t) - S(0) \tag{15}$$

along the chain by varying $r$, where $r = 0$ corresponds to the half-chain. Note that by subtracting the ground-state entropy $S(0)$, we aim to extract information about the excess entanglement created by a local excitation.

In the following subsections we first introduce an intuitive picture for the description of the entanglement spreading in terms of the low-lying quasiparticle excitations introduced in Sec. 2. We then proceed to the numerical study of the entanglement profiles after exciting the ground state with a fermionic creation operator, and compare the results to our quasiparticle ansatz. In the last part we consider an excitation created by a local Majorana fermion operator.

### 3.1 Entanglement spreading in the quasiparticle picture

Let us consider an excitation above the ground state of the XXZ chain by acting with a fermion creation operator $c_j^\dagger$. To capture the dynamics, one would have to first decompose the initial local excited state in the eigenbasis of the Hamiltonian. As discussed in the previous section, these eigenstates are described by quasiparticles parametrized by their rapidities or quasimomenta. The entanglement properties of various eigenstates in the XXZ chain were studied before in [37, 38], whereas a systematic CFT treatment of low-energy excitations was introduced in [39, 40]. In the framework of free quantum field theory, a surprisingly simple result on quasiparticle excitations was recently found in [41, 42]. Namely, the excess entanglement measured from the ground state was found to be completely independent of the quasiparticle momenta, depending only on the ratio $p$ of the subsystem and full chain lengths. Moreover, for quasiparticles described by a single momentum, the excess entropy is given by a binary formula $\Delta S = -p \ln p - (1-p) \ln(1-p)$, which allows for a simple probabilistic interpretation. Indeed, the ratio $p$ is just the probability of finding the quasiparticle within the subsystem.

Motivated by these results, we now put forward a simple ansatz for the spreading of entanglement after the local excitation. Under time evolution, the quasiparticles involved in the decomposition of the initial state spread out with their corresponding group velocities. However, our main assumption is that their contribution to entanglement is still independent of the momentum. Furthermore, we shall also assume that the dominant part of the entanglement is carried by the lowest-lying spinon modes, and that a spatially localized excitation translates to a homogeneous distribution of the momenta in the initial state. Under these assumptions we expect that the entanglement profile at time $t \gg 1$

and distance $r \gg 1$ from the excitation, in the space-time scaling limit $\zeta = r/t$ fixed, is determined exclusively via

$$\mathcal{N} = \int_0^\pi \frac{\mathrm{d}q}{\pi} \, \Theta(v_s(q) - \zeta) \,, \tag{16}$$

where $\Theta(x)$ is the Heaviside step function and $v_s(q)$ is the spinon velocity. In fact, this is nothing else but the fraction of the spinon modes with sufficient velocity to arrive at the subsystem. The simple probabilistic interpretation of the entanglement then leads to the binary entropy formula for the profile

$$\Delta S = -\mathcal{N} \ln(\mathcal{N}) - (1 - \mathcal{N}) \ln(1 - \mathcal{N}) \,. \tag{17}$$

In particular, for the gapless case considered here, inserting the expression (5) of the spinon velocity into (16), the spinon fraction can immediately be found as

$$\mathcal{N} = \frac{1}{\pi} \arccos\left(\frac{\zeta}{v}\right) \,, \tag{18}$$

where $v = v_s(0)$ denotes the maximal spinon velocity.

In summary, our simplistic ansatz (17) provides an interpretation of the excess entropy based on the dispersive dynamics of the quasiparticle modes, where $\mathcal{N}$ is the fraction of the initially localized excitation that arrives at the subsystem. In fact, the very same ansatz has recently been suggested for the description of entanglement spreading after local fermionic excitations in the XY chain, finding an excellent agreement with numerics [28]. Note, however, that the XY chain is equivalent to a free-fermion model and thus all the single-particle modes can exactly be included in $\mathcal{N}$. In contrast, for the interacting XXZ chain, restricting ourselves to the spinon modes should necessarily introduce some limitations to the quasiparticle ansatz, as demonstrated in the following subsection.

## 3.2 Local fermionic excitation

We continue with the numerical study of the excitation produced by the fermionic creation operator $c_j^\dagger$. The fermion operators are related to the spin variables via the Jordan-Wigner transformation

$$c_j^\dagger = \left(\prod_{l=-L/2+1}^{j-1} \sigma_l^z\right) \sigma_j^+ \,, \qquad c_j = \left(\prod_{l=-L/2+1}^{j-1} \sigma_l^z\right) \sigma_j^- \,, \tag{19}$$

where $\sigma_j^\alpha$ are the Pauli matrices and $\sigma_j^\pm = \left(\sigma_j^x \pm i\sigma_j^y\right)/2$. For simplicity, we shall only consider the case where the excitation is created by $c_1^\dagger$ in the middle of the chain. The time-evolved state after the excitaiton is then given by

$$|\psi(t)\rangle = N^{-1/2} \mathrm{e}^{-iHt} c_1^\dagger |\psi_0\rangle \,, \tag{20}$$

where $|\psi_0\rangle$ is the ground state and the normalization is given by

$$N = \langle\psi_0| \, c_1 c_1^\dagger \, |\psi_0\rangle = 1/2 \tag{21}$$

as the ground state is half filled. The time evolution is actually implemented via time-dependent DMRG (tDMRG) [43, 44] in the spin-representation of the XXZ chain, by first carrying out the ground-state search and applying the string operator (19) onto the MPS representation of $|\psi_0\rangle$. The calculations were performed using the ITensor C++ library [45]

255  and a truncated weight of $10^{-9}$.

256

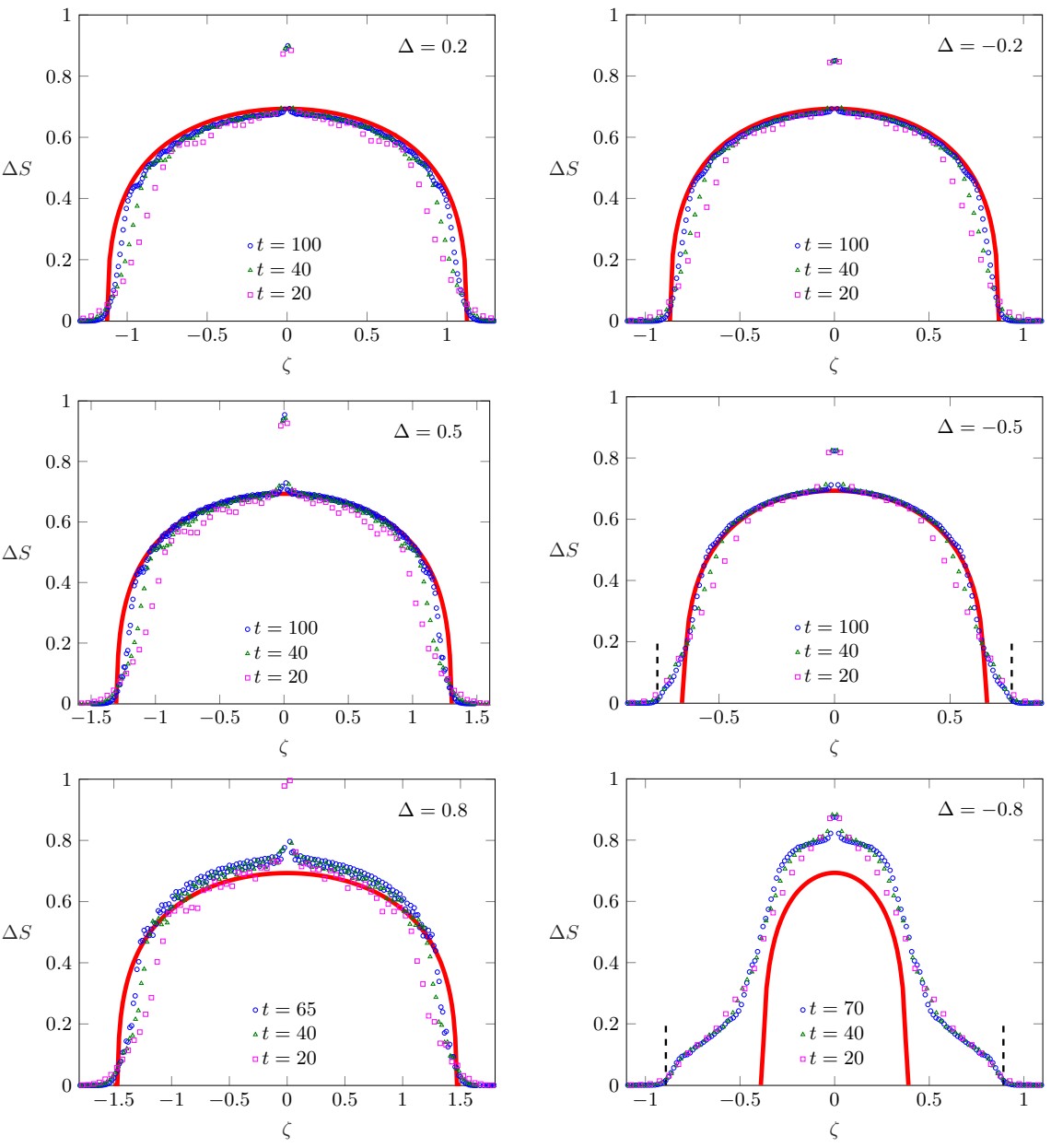

Figure 1: Excess entropy profiles $\Delta S$ obtained from tDMRG simulations at different times (symbols), after the excitation $c_1^{\dagger}$ in a chain of length $L = 300$. The scaled profiles are plotted against $\zeta = r/t$ and compared to the quasiparticle ansatz (red lines) in Eq. (17). The dashed black lines denote the maximum velocity of the particle-hole excitations, derived from Eq. (6).

257  The results of our simulations are shown in Fig. 1 for various interaction strengths
258  $\Delta$. The different symbols correspond to snapshots of the entropy profile $\Delta S$ at different
259  times, plotted against the scaled distance $\zeta = r/t$. The quasiparticle ansatz (17) computed
260  using the spinon fraction (18) is shown by the red solid lines. For moderate values of $|\Delta|$,
261  one observes a very good agreement with the numerical profiles, except for a peak around
262  $\zeta = 0$. Note that this peak rises above the maximum value $\ln(2)$ that can be obtained

from the spinon ansatz. A closer inspection for $r = 0$ indicates that the entropy peak also converges to a finite value for large times, with a nontrivial dependence on $\Delta$. Moreover, one can also observe a slight broadening of the peak for larger $\Delta$. However, the precise origin of the peak cannot be understood within our simple quasiparticle ansatz.

Systematic deviations from (17) also occur for larger $\Delta$, especially in the attractive regime. Indeed, for $\Delta = -0.5$ one already observes that the edges of the profile obtained from numerics fall slightly outside of the spinon edge, whereas the bulk profile still shows a good agreement. For $\Delta = -0.8$ the mismatch becomes more drastic both in the bulk and around the edges, signaling the breakdown of the naive spinon ansatz. Clearly, for strong attractive interactions the local excited state should have significant overlaps with other quasiparticle excitations of the XXZ chain. In fact, as discussed in Sec. 2, in this regime the maximum velocity of particle-hole excitations exceeds the spinon velocity and matches perfectly the edges of the profile, as indicated by the black dashed lines in Fig. 1. Hence, the entropy spreading should be determined by the coexistence of the spinon and particle-hole excitations, allowing to reach values beyond $\ln(2)$. Presumably, improving the ansatz (17) would require the knowledge of the overlaps with the different families of quasiparticles. Finally, it should be noted that, even though the edge locations of the profile seem to be captured, significant deviations in the bulk also occur for large repulsive interactions (see $\Delta = 0.8$ in Fig. 1), which might be due to bound-state contributions.

## 3.3 Local Majorana excitation

As a second example, we are going to consider local Majorana excitations, given in terms of the spin variables via

$$m_{2j-1} = \left(\prod_{l=-L/2+1}^{j-1} \sigma_l^z\right)\sigma_j^x, \qquad m_{2j} = \left(\prod_{l=-L/2+1}^{j-1} \sigma_l^z\right)\sigma_j^y, \tag{22}$$

and satisfying the anticommutation relations $\{m_k, m_l\} = 2\delta_{kl}$. Majorana operators are Hermitian and related to the fermion creation/annihilation operators as $m_{2j-1} = c_j + c_j^\dagger$ and $m_{2j} = i\left(c_j - c_j^\dagger\right)$. Focusing again on an excitation $m_1$ in the middle of the chain, the time-evolved stated is now given by

$$|\psi(t)\rangle = \mathrm{e}^{-iHt}m_1|\psi_0\rangle . \tag{23}$$

The entanglement profiles $\Delta S$ obtained from tDMRG simulations of (23) are depicted in Fig. 2 for four different values of $\Delta$. To visualize the spreading of the profile, we now plot the unscaled data against the location of the subsystem boundary. For $\Delta = 0$, the profile looks similar to that of the corresponding $c_1^\dagger$ excitation and is indeed perfectly reproduced by the quasiparticle ansatz (17). However, in the interacting case $\Delta \neq 0$, one observes a marked difference when compared to the corresponding panels in Fig. 1. Namely, the profiles in Fig. 2 clearly exceed the value $\ln(2)$, indicated by the dashed horizontal lines, which is the maximum of the ansatz (17). Nevertheless, we observe that the profiles after the $m_1^\dagger$ excitations can be well described by a simple rescaling of the spinon ansatz (17), as shown by the solid lines in Fig. 2. The constant factor multiplying the ansatz is chosen such that the maxima of the profiles at $r = 0$ are correctly reproduced. Note also that the central peak observed for the $c_1^\dagger$ excitation in Fig. 1 is missing for the Majorana excitation.

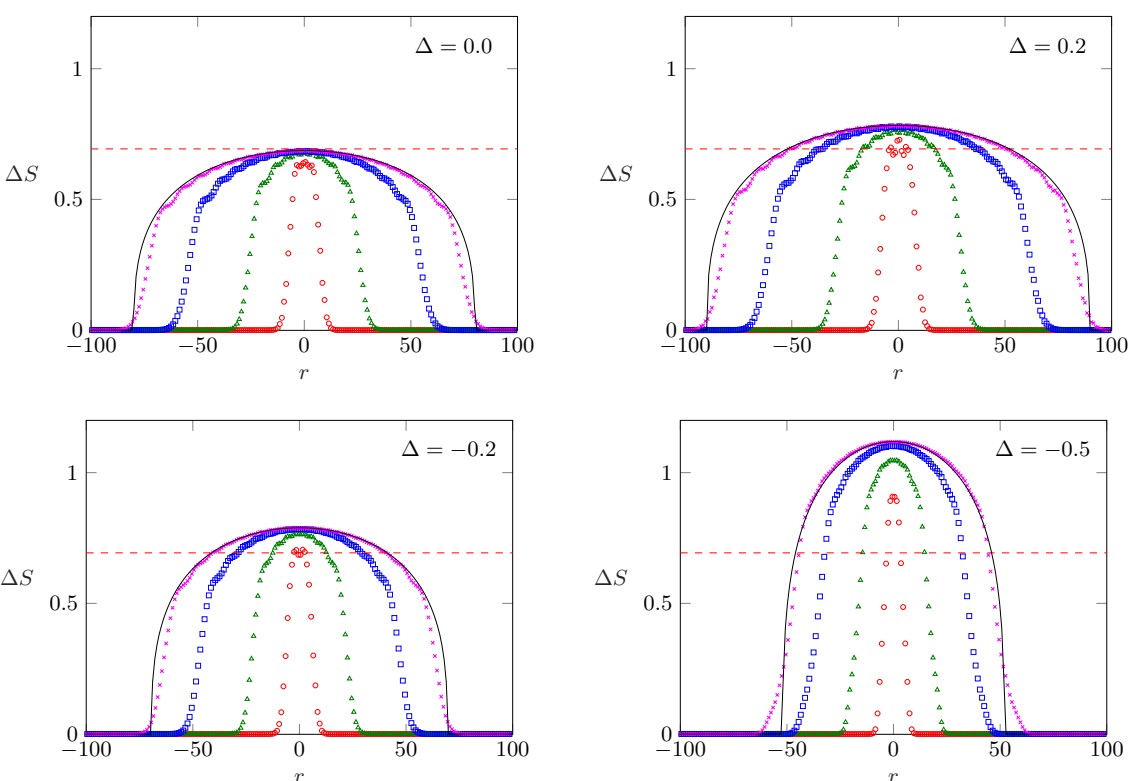

Figure 2: Excess entropy profiles $\Delta S$ as a function of $r$ at times $t = 10, 30, 60, 80$ (red, green, blue, magenta) after the Majorana excitation $m_1$ for four different values of $\Delta$ and $L = 200$. The red dashed lines indicate the value $\ln(2)$. The black solid lines show the spinon ansatz Eq. (17) for $t = 80$, multiplied by a constant to match the maxima of the profiles.

To better understand the behaviour of the maxima, on the left of Fig. 3 we plot the time evolution of the excess entropy $\Delta S$ in the middle of the chain ($r = 0$) with $L = 200$ and for various $\Delta$. One observes that the asymptotic value of the excess entropy grows with increasing $|\Delta|$, approaching its maximum very slowly in time. In fact, for even larger times the entropy starts to decrease again as one approaches $vt \approx L$, when the fastest spinons leave the subsystem after a reflection from the chain end. This is demonstrated on the right of Fig. 3 by repeating the calculations for a smaller chain with $L = 50$. The emergence of a plateau is clearly visible, which then immediately repeats itself for $vt > L$ due to the symmetry of the geometry, with the spinons reflected from the other end of the chain entering the subsystem again. However, the question why the height of the plateau depends on the interaction strength $\Delta$ can only be answered via a more involved CFT analysis of the problem, which is presented in the next section.

## 4 Entanglement after local excitations in CFT

The low-energy physics of the gapless XXZ chain can be captured within quantum field theory via the bosonization procedure [46]. Using the fermionic representation (2) of the chain, one introduces the Heisenberg operators $c(x, \tau) = e^{\tau H} c_x e^{-\tau H}$, where $x$ is the spatial coordinate along the chain and we introduced the imaginary time $\tau = it$. Linearizing the dispersion around the Fermi points, one can approximate

$$c(x, \tau) \simeq e^{ik_F x} \psi(x, \tau) + e^{-ik_F x} \bar{\psi}(x, \tau), \tag{24}$$

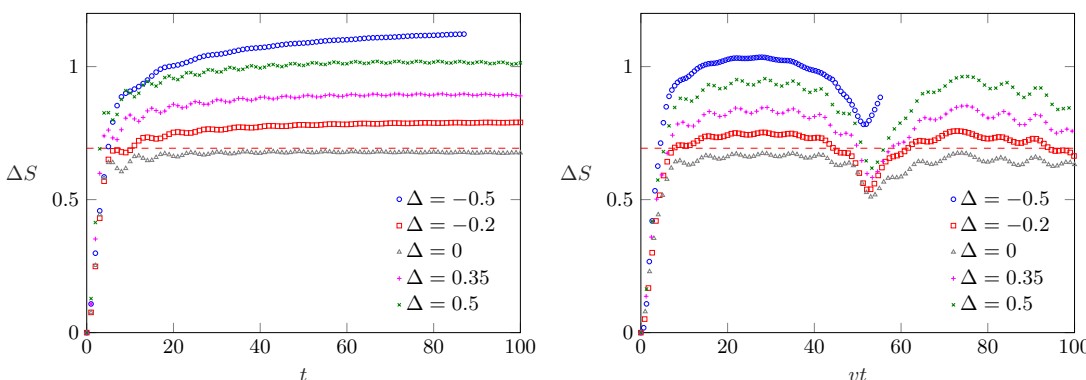

Figure 3: Left: Entropy growth in the middle of the chain $r = 0$, after the Majorana excitation $m_1$ for different values of $\Delta$ and $L = 200$. The red dashed line indicates the value $\ln(2)$. Right: $\Delta S$ for a smaller chain with $L = 50$ against the scaled time $vt$ for the same $\Delta$ values.

where $\psi(x,\tau)$ and $\bar{\psi}(x,\tau)$ are the right and left-moving components of a fermion field. The phase factors with the Fermi momentum, where $k_F = \pi/2$ for a half-filled chain, are included to ensure that the chiral fermions are described by slowly varying fields. Introducing the complex coordinates $w = v\tau - ix$ and $\bar{w} = v\tau + ix$, where $v$ denotes the Fermi velocity, they can be written in a bosonized form [46]

$$\psi(w) = \frac{1}{\sqrt{2\pi}} e^{-i\sqrt{4\pi}\varphi(w)} , \qquad \bar{\psi}(\bar{w}) = \frac{1}{\sqrt{2\pi}} e^{i\sqrt{4\pi}\bar{\varphi}(\bar{w})} , \tag{25}$$

where $\varphi(w)$ and $\bar{\varphi}(\bar{w})$ are the chiral boson fields. In terms of the new bosonic variables

$$\phi = \varphi + \bar{\varphi} , \qquad \theta = \varphi - \bar{\varphi} , \tag{26}$$

one can show that the bosonized form of the XXZ chain (2) is described by the Luttinger liquid Hamiltonian [47]

$$H_{LL} = \frac{v}{2} \int \mathrm{d}x \left[ K(\partial_x \theta)^2 + K^{-1}(\partial_x \phi)^2 \right] . \tag{27}$$

Apart from the velocity $v$, the Hamiltonian (27) is characterized by the Luttinger parameter $K$. Both of them can be fixed from the exact Bethe ansatz solution as

$$v = \frac{\pi}{2} \frac{\sin(\gamma)}{\gamma} , \qquad K = \frac{1}{2} \left( 1 - \frac{\gamma}{\pi} \right)^{-1} , \tag{28}$$

with the usual parametrization $\Delta = \cos(\gamma)$. Note that $v = v_s(0)$ is just the maximum of the spinon velocity (5).

   In CFT language, the Luttinger liquid corresponds to a free compact boson field theory. In order to study entanglement evolution after local operator excitations, we shall thus use the framework developed for a generic CFT [15, 16]. In the following we summarize the main steps of the procedure. Let us consider the state

$$|\psi\rangle = N^{-1/2} \mathcal{O}(-d) |0\rangle \tag{29}$$

excited from the CFT vacuum $|0\rangle$ by insertion of the local operator $\mathcal{O}(-d)$, where $N$ accounts for the normalization of the state. For the sake of generality, we consider the

situation where the excitation is inserted at a distance $d$ measured from the center of the chain. After time evolution, the density matrix reads

$$\rho(t) = N^{-1} \mathrm{e}^{-iHt} \mathrm{e}^{-\epsilon H} \mathcal{O}(-d) |0\rangle \langle 0| \mathcal{O}^\dagger(-d) \mathrm{e}^{-\epsilon H} \mathrm{e}^{iHt}, \tag{30}$$

where $\epsilon$ is a UV regularization that is required for the state to be normalizable. Working in a Heisenberg picture, the time evolution can be absorbed into the operators, and the state can be represented as

$$\rho(t) = \frac{\mathcal{O}(w_2, \bar{w}_2) |0\rangle \langle 0| \mathcal{O}^\dagger(w_1, \bar{w}_1)}{\langle \mathcal{O}^\dagger(w_1, \bar{w}_1) \mathcal{O}(w_2, \bar{w}_2) \rangle}, \tag{31}$$

where the complex coordinates of the operator insertions are given by

$$\begin{aligned}
w_1 &= -i(vt - d) + \epsilon, & \bar{w}_1 &= -i(vt + d) + \epsilon, \\
w_2 &= -i(vt - d) - \epsilon, & \bar{w}_2 &= -i(vt + d) - \epsilon.
\end{aligned} \tag{32}$$

It should be stressed that the $\bar{w}_j$ coordinates are actually not the complex conjugates of $w_j$, as we are assuming $\tau = it$ to be real, such that we can work with Euclidean spacetime.

With the expression (31) at hand, one can proceed to construct the path-integral representation of the reduced density matrix, by opening a cut at $\tau = 0$ along the spatial coordinates of the subsystem $A$. The Rényi entropy

$$S_n(t) = \frac{1}{1-n} \ln \mathrm{Tr}\left[\rho_A^n(t)\right] \tag{33}$$

for integer $n$ can then be obtained by applying the replica trick [48], i.e. sewing together $n$ copies of the path integrals cyclically along the cuts. In turn, one can express the excess Rényi entropy $\Delta S_n = S_n(t) - S_n(0)$ via correlation functions of the local operator as [15, 16]

$$\Delta S_n = \frac{1}{1-n} \log \left[ \frac{\langle \mathcal{O}^\dagger(w_1, \bar{w}_1) \mathcal{O}(w_2, \bar{w}_2) \ldots \mathcal{O}(w_{2n} \bar{w}_{2n}) \rangle_{\Sigma_n}}{\langle \mathcal{O}^\dagger(w_1, \bar{w}_1) \mathcal{O}(w_2, \bar{w}_2) \rangle_{\Sigma_1}^n} \right], \tag{34}$$

where $\Sigma_n$ denotes the $n$-sheeted Riemann surface, with $w_1, \ldots, w_{2n}$ and $\bar{w}_1, \ldots, \bar{w}_{2n}$ being the replica coordinates of the insertion points (32).

Although the expression (34) for the excess Rényi entropy is very general, the calculation of $2n$-point functions on the complicated Riemann surface $\Sigma_n$ may become rather involved. However, if the subsystem $A$ is given by a single interval $0 \leq x \leq \ell$ in an infinite chain, the geometry can be simplified by the conformal transformation

$$z = \left( \frac{w}{w + i\ell} \right)^{1/n}, \qquad \bar{z} = \left( \frac{\bar{w}}{\bar{w} - i\ell} \right)^{1/n}, \tag{35}$$

which maps the $n$-sheeted surface onto a single Riemann sheet. This transformation leads to the holomorphic coordinates of the operator insertions

$$z_{2j-1} = e^{2\pi ij/n} \left( \frac{d - vt - i\epsilon}{\ell + d - vt - i\epsilon} \right)^{1/n}, \qquad z_{2j} = e^{2\pi ij/n} \left( \frac{d - vt + i\epsilon}{\ell + d - vt + i\epsilon} \right)^{1/n}, \tag{36}$$

while the anti-holomorphic ones are given by

$$\bar{z}_{2j-1} = e^{-2\pi ij/n} \left( \frac{d + vt + i\epsilon}{\ell + d + vt + i\epsilon} \right)^{1/n}, \qquad \bar{z}_{2j} = e^{-2\pi ij/n} \left( \frac{d + vt - i\epsilon}{\ell + d + vt - i\epsilon} \right)^{1/n}. \tag{37}$$

Furthermore, if the local operators are primary fields of the CFT with respective conformal dimensions $h_{\mathcal{O}}$ and $\bar{h}_{\mathcal{O}}$, the $2n$-point function transforms as

$$\langle\prod_{j=1}^{n}\mathcal{O}^{\dagger}(w_{2j-1},\bar{w}_{2j-1})\mathcal{O}(w_{2j},\bar{w}_{2j})\rangle_{\Sigma_n}=\prod_{i=1}^{2n}\left(\frac{\mathrm{d}w}{\mathrm{d}z}\right)_{z_i}^{-h_{\mathcal{O}}}\left(\frac{\mathrm{d}\bar{w}}{\mathrm{d}\bar{z}}\right)_{\bar{z}_i}^{-\bar{h}_{\mathcal{O}}}\langle\prod_{j=1}^{n}\mathcal{O}^{\dagger}(z_{2j-1},\bar{z}_{2j-1})\mathcal{O}(z_{2j},\bar{z}_{2j})\rangle_{\Sigma_1}.$$

$$(38)$$

In the end, one is left with a problem of calculating $2n$-point functions on the complex plane. For the sake of simplicity, in the following we shall only consider the case $n = 2$, and apply the procedure outlined above to the Luttinger liquid theory, with the local excitations considered in section 3.

## 4.1 Fermionic excitation

We start with the fermion creation operator, which after bosonization (25) corresponds to the field insertion

$$\mathcal{O}_f(w,\bar{w})=\mathrm{e}^{ik_Fd}\mathrm{e}^{i\sqrt{4\pi}\varphi(w)}+\mathrm{e}^{-ik_Fd}\mathrm{e}^{-i\sqrt{4\pi}\bar{\varphi}(\bar{w})}\,,\tag{39}$$

where we omitted normalization factors that cancel in the expression (34). Clearly, $\mathcal{O}_f(w,\bar{w})$ is not itself a primary operator but rather a linear combination of two. Hence, the calculation of the four-point function that appears in $\Delta S_2$ involves a number of terms with primaries, each of which can be mapped from $\Sigma_2$ to the complex plane using the transformation rule (38). The calculation of these correlation functions can be facilitated by first performing a canonical transformation

$$\theta'=\sqrt{K}\theta\,,\qquad\phi'=\frac{1}{\sqrt{K}}\phi\,.\tag{40}$$

which absorbs the Luttinger parameter $K$ in the Hamiltonian (27). However, since the variables $\theta$ and $\phi$ are actually linear combinations (26) of the chiral bosons, the change of variables corresponds to the Bogoliubov transformation

$$\varphi=\cosh(\xi)\varphi'+\sinh(\xi)\bar{\varphi}'\qquad\bar{\varphi}=\sinh(\xi)\varphi'+\cosh(\xi)\bar{\varphi}'\,,\tag{41}$$

where $K=\mathrm{e}^{2\xi}$. Thus, the transformation of the Luttinger liquid Hamiltonian induces a left-right mixing of the chiral bosonic modes. In the following we shall use the shorthand notations $c=\cosh(\xi)$ and $s=\sinh(\xi)$.

Clearly, our task now boils down to evaluate correlation functions of vertex operators

$$V_{\alpha,\beta}(z,\bar{z})=\mathrm{e}^{i\sqrt{4\pi}\alpha\varphi'(z)+i\sqrt{4\pi}\beta\bar{\varphi}'(\bar{z})}\tag{42}$$

on the complex plane with respect to the Luttinger liquid theory scaled to the free-fermion point. The $n$-point function of vertex operators is then well known and given by [49]

$$\langle\prod_{j=1}^{n}V_{\alpha_i,\beta_i}(z_i,\bar{z}_i)\rangle=\prod_{i<j}(z_i-z_j)^{\alpha_i\alpha_j}(\bar{z}_i-\bar{z}_j)^{\beta_i\beta_j}\,,\tag{43}$$

where the neutrality conditions

$$\sum_{i=1}^{n}\alpha_i=0\,,\qquad\sum_{i=1}^{n}\beta_i=0\tag{44}$$

must be satisfied, otherwise the correlator vanishes. In particular, considering the two-point function one immediately sees that the vertex operator (42) is a primary with scaling dimensions $h=\alpha^2/2$ and $\bar{h}=\beta^2/2$.

With all the ingredients at hand, performing the calculation for $\Delta S_2$ is a straightforward but cumbersome exercise, and we refer to Appendix A for the main details. It turns out that the result depends only on the cross-ratios

$$\eta = \frac{z_{12} z_{34}}{z_{13} z_{24}}, \qquad \bar{\eta} = \frac{\bar{z}_{12} \bar{z}_{34}}{\bar{z}_{13} \bar{z}_{24}} \tag{45}$$

of the holomorphic and anti-holomorhic coordinates (36) and (37), where $z_{ij} = z_i - z_j$ and $\bar{z}_{ij} = \bar{z}_i - \bar{z}_j$, respectively. In terms of the cross-ratios, the final result reads

$$\Delta S_2 = -\ln\left( \frac{1 + |\eta|^{(c+s)^2} + |1-\eta|^{(c+s)^2}}{2} \right). \tag{46}$$

It is important to stress that the notation $|\eta|$ should be understood as $(\eta\bar{\eta})^{1/2}$, since the two cross ratios are not conjugate variables. In particular, in the limit $\epsilon \to 0$ of the regularization, one has the behaviour [15, 16]

$$\lim_{\epsilon \to 0} \eta = \begin{cases} 0 & \text{if } 0 < vt < d \text{ or } vt > d + \ell \\ 1 & \text{if } d < vt < d + \ell \end{cases}, \qquad \lim_{\epsilon \to 0} \bar{\eta} = 0. \tag{47}$$

This yields the following limit for the Rényi entropy

$$\lim_{\epsilon \to 0} \Delta S_2 = \begin{cases} 0 & \text{if } 0 < vt < d \text{ and } vt > d + \ell \\ \ln(2) & \text{if } d < vt < d + \ell \end{cases}. \tag{48}$$

The result has a very simple interpretation. Namely, our excitation is an equal superposition of a left- and right-moving fermion, and the entanglement is changed by $\ln(2)$ only when the right-moving excitation is located within the interval. In fact, this is exactly the same picture that lies behind the quasiparticle ansatz (17), without the dispersion of the wavefront. Interestingly, apart from the presence of the spinon velocity $v$, the limiting result (48) is independent of the anisotropy $\Delta$. The only effect of the left-right boson mixing appears in the exponents of the cross-ratios in (46), which simply determines how the sharp step-function for $\Delta S_2$ is rounded off for finite UV regularizations. In fact, this result is very similar to the one obtained for a non-chiral EPR-primary excitation in Ref. [16, 19]. Moreover, this is also a simple generalization of the result in Ref. [25], where the superposition of purely holomorphic and anti-holomorphic primaries was considered.

## 4.2 Majorana excitation

We move on to consider the Majorana excitation

$$\mathcal{O}_m(w, \bar{w}) = \mathcal{O}_f(w, \bar{w}) + \mathcal{O}_f^\dagger(w, \bar{w}). \tag{49}$$

The calculation of $\Delta S_2$ follows the exact same procedure as for $\mathcal{O}_f(w, \bar{w})$, however, one has now an even larger number of terms to consider. The main steps are again outlined in Appendix A, which lead to the result

$$\Delta S_2 = -\ln\left( \frac{2A + B + C}{8} \right), \tag{50}$$

where the terms in the logarithm are given by

$$A = |1 - \eta|^{(c+s)^2} + |1 - \eta|^{(c-s)^2} + |\eta|^{(c+s)^2} + |\eta|^{(c-s)^2} \tag{51}$$

$$B = 2 + \eta^{2c^2} \bar{\eta}^{2s^2} + \eta^{2s^2} \bar{\eta}^{2c^2} + (1 - \eta)^{2c^2} (1 - \bar{\eta})^{2s^2} + (1 - \eta)^{2s^2} (1 - \bar{\eta})^{2c^2} \tag{52}$$

$$C = \left[ |\eta|^{(c+s)^2} |1 - \eta|^{(c-s)^2} + |\eta|^{(c-s)^2} |1 - \eta|^{(c+s)^2} \right] (Z + \bar{Z}) \tag{53}$$

and a new variable is introduced as

$$Z = \frac{z_1 \bar{z}_2 (1 - \bar{z}_1^2)(1 - z_2^2)}{\bar{z}_1 z_2 (1 - z_1^2)(1 - \bar{z}_2^2)} \,. \tag{54}$$

The result is thus rather involved and cannot be written as a function of the cross-ratios alone. However, in the limit $\epsilon \to 0$, the factors in $A$, $B$, and $C$ can trivially be evaluated using (47), as well as using $Z \to 1$ and $\bar{Z} \to 1$. For the case $\Delta \neq 0$, this leads to the following simple result

$$\lim_{\epsilon \to 0} \Delta S_2 = \begin{cases} 0 & \text{if } 0 < vt < d \text{ and } vt > d + \ell \\ 2\ln(2) & \text{if } d < vt < d + \ell \end{cases} \,. \tag{55}$$

In sharp contrast, for $\Delta = 0$, where $c = 1$ and $s = 0$, one recovers the result (48). Hence, one arrives at the rather surprising result that the excess entropy is doubled in case of interactions, which must be a consequence of the left-right boson mixing.

Obviously, for finite values of the regularization $\epsilon$, this transition should take place continuously, rather than giving an abrupt jump. The behaviour of $\Delta S_2$ for $\epsilon = 0.1$ is shown in Fig. 4 for an interval of length $\ell = 20$ at a distance $d = 10$ from the excitation. One can clearly see the development of a plateau for times $d < vt < d + \ell$, the height of which increases monotonously with $\Delta$. Nevertheless, even for the largest value $\Delta = 0.8$, the expected maximum of $2\ln(2)$ is by far not reached. The very slow convergence towards the $\epsilon \to 0$ (or, equivalently, $t \to \infty$) limit can be understood by looking at the structure of the terms appearing in (50). In fact, for smaller values of $|\Delta|$, the slowest converging pieces are given by $\eta^{2c^2} \bar{\eta}^{2s^2}$ as well as $(1 - \eta)^{2s^2}(1 - \bar{\eta})^{2c^2}$ in the expression (52) of $B$, due to the large-time behaviour $\bar{\eta} \approx 1 - \eta \approx (\epsilon/2vt)^2$ for $d \ll vt \ll \ell + d$. Hence, the apparent nontrivial values of the plateau in Fig. 4 is a consequence of the very slow decay $(\epsilon/vt)^{4s^2}$, where the exponent for e.g. $\Delta = 0.5$ is given by $4s^2 \approx 0.08$. Clearly, observing convergence towards $\Delta S_2 \to 2\ln(2)$ would require enormous time scales as well as interval lengths.

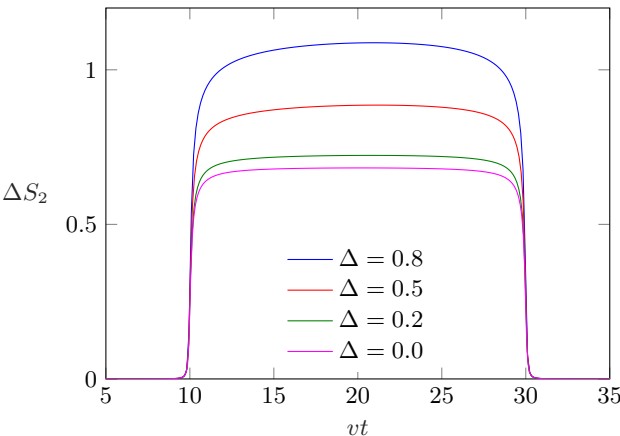

Figure 4: Time evolution of the excess Rényi entropy in Eq. (50) after the Majorana excitation with $\ell = 20$, $d = 10$ and $\epsilon = 0.1$.

Despite the different geometry considered for the CFT calculations, we expect that the result (50) should also give quantitative predictions for the finite XXZ chain in a certain regime. First of all, for the half-chain bipartition where the excitation is applied directly at the boundary, the role of the dispersion should not play an important role, as all the excitations can immediately enter the subsystem. Furthermore, one could argue that the

finite chain effectively corresponds to an interval of size $\ell = L$, which is the distance the quasiparticles have to cover before leaving the subsystem after reflection from the chain end. Clearly, the exact form of the plateau will not be the same in the two cases, but one expects the CFT results to be applicable in a regime $vt \ll L$. Finally, there is a highly nontrivial symmetry $s \to -s$ displayed by all the terms (51)-(53) in the expression of $\Delta S_2$, corresponding to a change of the Luttinger parameter $K \to 1/K$, which is expected to be observed also in the lattice calculations. Note that since $K = 1$ corresponds to the free-fermion point $\Delta = 0$, the symmetry relates interaction strengths of different sign.

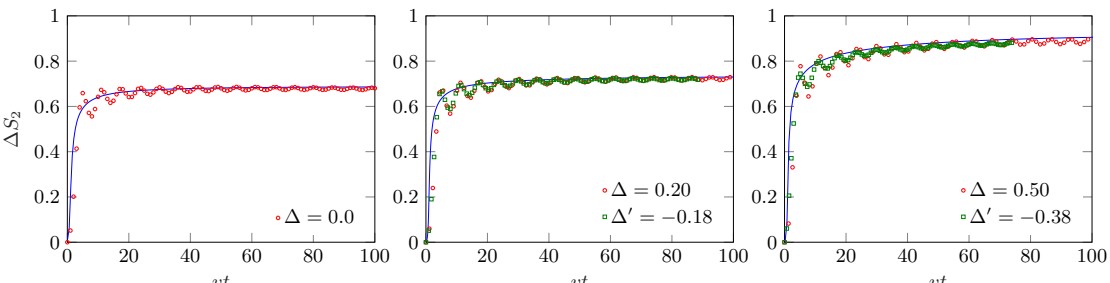

Figure 5: Growth of the Rényi entropy $\Delta S_2$ for pairs of conjugate interaction parameters $\Delta$ and $\Delta'$ (red and green symbols) for a chain of length $L = 200$. The blue solid lines show the CFT result Eq. (50) with $\ell = 200$ and $d = 1$. The regularization $\epsilon = 0.55, 0.40, 0.35$ (from left to right) was tuned to obtain the best match with the tDMRG data.

In Fig. 5 we show a comparison of $\Delta S_2$ obtained from tDMRG calculations for a XXZ chain with $L = 200$ divided in the middle, to the CFT result (50) shown by the blue solid lines. For the latter we have set $\ell = L$ and $d = 1$ as discussed above, whereas the regularization $\epsilon$ was set by hand in order to achieve the best agreement with the numerical data. One indeed observes that the CFT result gives, up to oscillations, a good quantitative description of the XXZ numerics. Furthermore, for each $\Delta \neq 0$, we also performed the calculation for the conjugate $\Delta'$ corresponding to $K' = 1/K$, leading to a remarkably good collapse of the curves.

## 5 Entanglement dynamics in the gapped phase

The CFT studies of the previous section give a rather good qualitative description of the entanglement spreading in the critical phase of the XXZ chain. To obtain a complete picture, in this section we shall study the dynamics in the gapped antiferromagnetic phase. For a physically motivated setting, we choose one of the symmetry-broken ground states $|\psi_\uparrow\rangle$ from Eq. (8), with a nonvanishing staggered magnetization (9). We now consider local Majorana operators, defined in terms of the spin variables as

$$\tilde{m}_{2j-1} = \left( \prod_{l=-L/2+1}^{j-1} \sigma_l^x \right) \sigma_j^z , \qquad \tilde{m}_{2j} = \left( \prod_{l=-L/2+1}^{j-1} \sigma_l^x \right) \sigma_j^y . \tag{56}$$

Note that these operators differ from the ones in (22) discussed in the gapless phase by an interchange of the $x$ and $z$ spin components, but they also obey Majorana fermion statistics with anticommutation relations $\{\tilde{m}_k, \tilde{m}_l\} = 2\delta_{kl}$. We focus on the case of a domain wall created by $\tilde{m}_1$ in the center of the chain, which is then time evolved by the XXZ Hamiltonian (1)

$$|\psi(t)\rangle = \mathrm{e}^{-iHt}\tilde{m}_1 |\psi_\uparrow\rangle . \tag{57}$$

Note that, in order to find the proper symmetry-broken ground state, in the DMRG simulation we add to the Hamiltonian a small staggered field in the $z$-direction, which is then decreased towards zero during the sweeps.

First we have a look at the entropy growth $\Delta S$ for the half-chain $r = 0$ as a function of time, shown on the left of Fig. 6 for several values of the anisotropy $\Delta > 1$. One observes a clear saturation of the excess entropy for large times, which is reached very quickly for large values of $\Delta$. The asymptotic value of $\Delta S$ decreases with $\Delta$ and always exceeds $\ln(2)$. Remarkably, as shown on the right of Fig. 6, we find that the asymptotic excess entropy is well described by the formula $\Delta S = S(0) + \ln(2)$, where $S(0)$ is the ground-state entropy of the half-chain in the symmetry-broken state. Repeating the calculation for the excess Rényi entropy $\Delta S_2$, we find the exact same relation with $S_2(0)$.

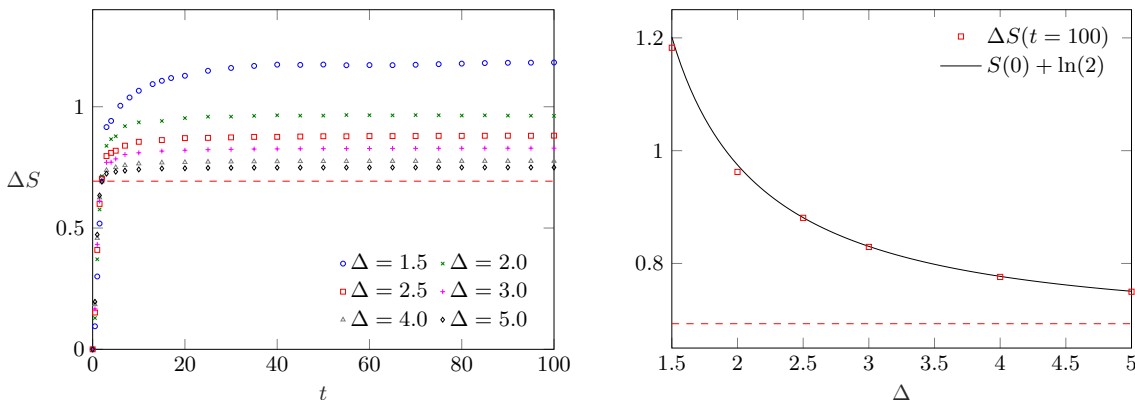

Figure 6: Left: Entanglement growth in the middle of the chain after a domain-wall excitation $\tilde{m}_1$ for different values of $\Delta > 1$ and $L = 400$. Right: $\Delta S$ at $t = 100$ compared to $S(0) + \ln(2)$ from Eq. (59). The red dashed line denotes $\ln(2)$. Note the different vertical scales.

To gain a deeper understanding of the above relation, one should invoke the exact results for the reduced density matrix of the half-chain, which can can be found with the corner transfer matrix (CTM) method as [50]

$$\rho_A = \frac{\mathrm{e}^{-H_{CTM}}}{\mathrm{Tr}\left(\mathrm{e}^{-H_{CTM}}\right)}, \qquad H_{CTM} = \sum_{j=0}^{\infty} \epsilon_j n_j, \tag{58}$$

where the single-particle eigenvalues are given by $\epsilon_j = 2j\phi$ with $\phi = \mathrm{acosh}(\Delta)$, and $n_j = 0, 1$ denotes fermionic occupation numbers. In other words, the entanglement Hamiltonian $H_{CTM}$ of the ground state is characterized by an equispaced single-particle entanglement spectrum. Strictly speaking, this result applies to a half-infinite chain, but in practice it holds also for finite chains of length much larger than the correlation length. Note also, that the result (58) applies for the symmetric ground state, whereas for the symmetry-broken state the term $j = 0$ is missing from the sum. In that case, the von Neumann and Rényi entropies can be simply expressed as [51]

$$S(0) = \sum_{j=1}^{\infty} \left[ \log\left(1 + \mathrm{e}^{-2j\phi}\right) + \frac{2j\phi}{1 + \mathrm{e}^{2j\phi}} \right], \tag{59}$$

as well as

$$S_n(0) = \frac{1}{1-n} \left[ \sum_{j=1}^{\infty} \log\left(1 + \mathrm{e}^{-2nj\phi}\right) - n \sum_{j=1}^{\infty} \log\left(1 + \mathrm{e}^{-2j\phi}\right) \right]. \tag{60}$$

It is easy to see that the inclusion of the term $j = 0$ with $\epsilon_0 = 0$ simply yields an extra $\ln(2)$ contribution to the entropies. This change alone, however, would not explain our findings for the asymptotic excess entropy in Fig. 6, which seems to indicate that $S(t) \approx 2S(0) + \ln(2)$ for $t \gg 1$. Indeed, in order to obtain such a formula, one would have to add a double degeneracy for each $\epsilon_j$ with $j \neq 0$. Let us now discuss how such a degeneracy is reflected in the eigenvalues $\lambda_l$ of the reduced density matrix. In fact, it is more convenient to introduce the scaled quantity

$$\nu_l = -\frac{1}{\phi}\ln\left(\frac{\lambda_l}{\lambda_0}\right), \tag{61}$$

where $\lambda_0$ denotes the maximal eigenvalue. For the initial symmetry-broken ground state, $\nu_l$ are independent of $\Delta$ and can only assume even integer values, with occasional multiplicities due to different integer partitions. The lowest lying $\lambda_l$ yield $\nu_l = 0, 2, 4, 6, 6, \ldots$, i.e. the first degeneracy appears as $6 = 2 + 4$. The inclusion of the $\epsilon_0 = 0$ term simply gives an overall double degeneracy of the levels $\lambda_l$. The doubling of the $\epsilon_j$ for $j \neq 0$ further increases the degeneracies. Altogether, the combined effect would lead to the multiplicities $(2, 4, 6)$ for $\nu_l = 0, 2, 4$.

To check these predictions, in Fig. 7 we have plotted the 12 lowest lying $\nu_l$ calculated from the reduced density matrix eigenvalues, as obtained from tDMRG simulations after time evolving the state (57) to $t = 100$. One can see that the $\nu_l$ lie indeed rather close to the expected even integer values, approximately reproducing the expected multiplicity structure. Interestingly, the largest deviation around $\nu_l = 4$ is found for $\Delta = 5$, where one actually finds the best agreement with the entropy formula, see Fig. 6. In fact, however, the contribution of these eigenvalues to the entropy is already negligible. Note that the situation for larger values of $\nu_l$ is much less clear, as they correspond to very small eigenvalues $\lambda_l$ which are already seriously affected by tDMRG truncation errors.

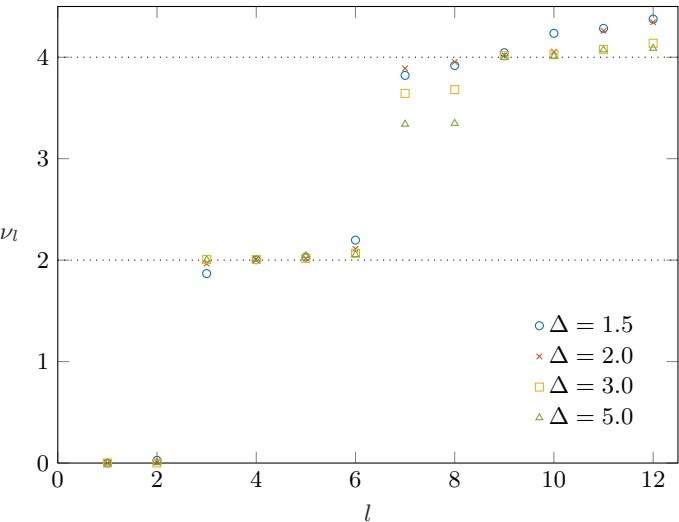

Figure 7: Scaled levels $\nu_l$ obtained from the reduced density matrix eigenvalues $\lambda_l$ at time $t = 100$ via Eq. (61) for different $\Delta$.

Although we find a nontrivial asymptotic behaviour of the half-chain entanglement, we expect that the full profile should still be described, up to a multiplicative factor, by the quasiparticle ansatz introduced in section 3.1, similarly to the Majorana excitation in

519 the gapless phase in Fig. 2. Therefore, we put forward the ansatz

$$\Delta S = \left(1 + \frac{S(0)}{\ln 2}\right) \left[-\mathcal{N} \ln\left(\mathcal{N}\right) - (1 - \mathcal{N}) \ln\left(1 - \mathcal{N}\right)\right], \tag{62}$$

520 and for the excess Rényi entropy we propose

$$\Delta S_n = \left(1 + \frac{S_n(0)}{\ln 2}\right) \frac{1}{1 - n} \ln\left[\mathcal{N}^n + (1 - \mathcal{N})^n\right]. \tag{63}$$

521 The quasiparticle fraction $\mathcal{N}$ must now be evaluated via (16) by using the spinon velocities
522 (13) in the gapped phase. Note that the binary entropy functions are multiplied by a factor
523 to reproduce our findings for the half-chain, where $\mathcal{N} = 1/2$. The results of our numerical
524 calculations for the profiles $\Delta S$ and $\Delta S_2$, plotted against the scaling variable $\zeta = r/t$, are
525 shown in Fig. 8. The solid lines show the respective ansatz (62) and (63), which give a
526 very good description of the data for both $\Delta$ values shown. In fact, we checked that the
527 profiles are nicely reproduced even for $\Delta = 1.5$, which already corresponds to a relatively
528 large correlation length.

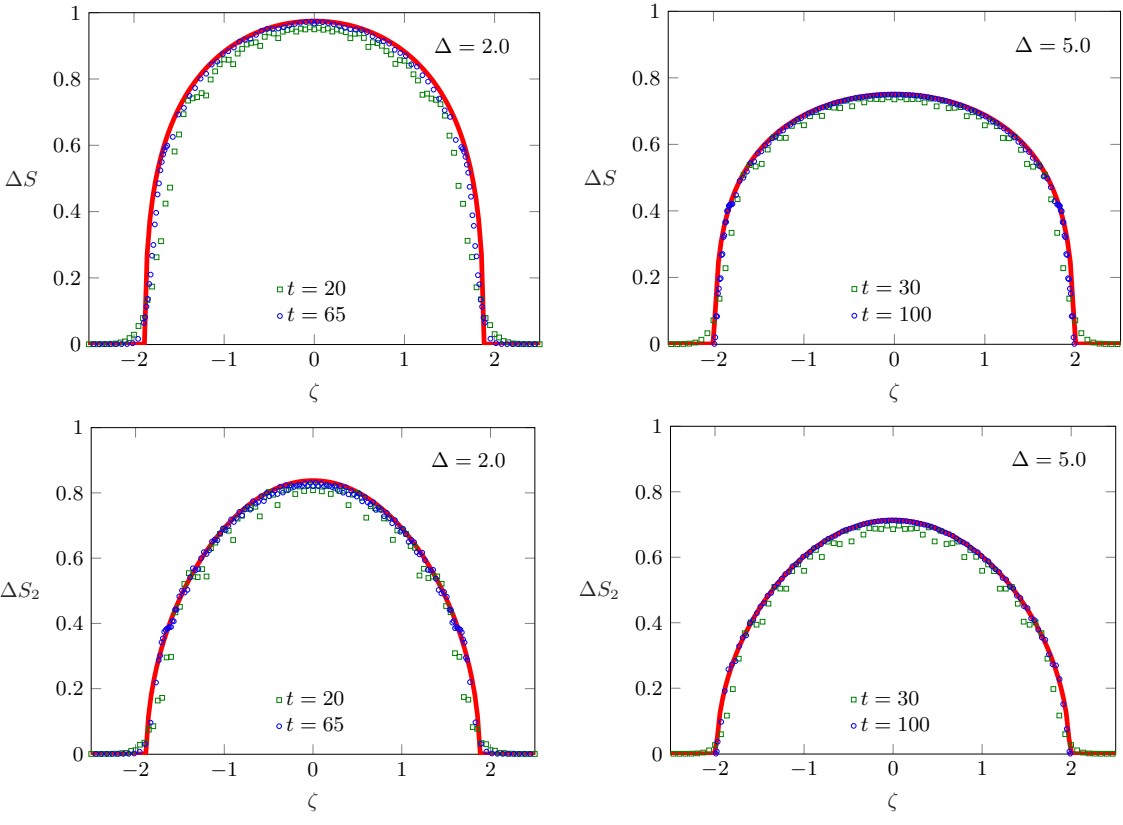

Figure 8: Entropy profiles $\Delta S$ (top) and $\Delta S_2$ (bottom) after a domain-wall excitation $\tilde{m}_1$ for two different value of $\Delta$ and $L = 400$. The solid lines show the ansatz Eq. (62) for the von Neumann, as well as Eq. (63) for the $n = 2$ Rényi excess entropy.

## 5.1 Magnetization profiles

530 To conclude this section, we also investigate the spreading of the magnetization profiles
531 for the antiferromagnetic domain wall excited by $\tilde{m}_1$. This setting was studied previously
532 with a focus on the edge behaviour of the profile [52]. In order to remove the dependence

on the ground-state value (9) of the staggered magnetization, we consider the normalized profile

$$\mathcal{M}_j(t) = \frac{\langle \psi(t)|\, \sigma_j^z\, |\psi(t)\rangle}{\langle \psi_\uparrow|\, \sigma_j^z\, |\psi_\uparrow\rangle}, \tag{64}$$

which then varies between $-1 \le \mathcal{M}_j(t) \le 1$ along the chain. We are mainly interested in the quasiparticle description of the time-evolved profile. In fact, a very similar problem was studied for a ferromagnetic domain wall in the XY chain [28], by first expanding the excited state in the single-particle basis of the Hamiltonian, which can then be time evolved trivially.

Here we assume that the dominant weight for our simple domain wall is carried by single-spinon excitations $|q\rangle$. Strictly speaking, this is only possible if one considers antiperiodic or open boundary conditions on the spins, since for a periodic chain spinons are created in pairs (i.e. one actually has a pair of domain walls). The time evolved state can then be written as

$$|\psi(t)\rangle \simeq \sum_q \mathrm{e}^{-it\varepsilon_s(q)} c(q)\, |q\rangle\ , \tag{65}$$

where $\varepsilon_s(q)$ is the spinon dispersion (10), while $c(q)$ are the overlaps of the domain-wall excitation with the single-spinon states. Note that the momentum of a single spinon satisfies $0 \le q \le \pi$, however, the total momentum of spinons above the quasidegenerate ground state is shifted by $\pi$. Since the domain wall is created by a strictly local fermionic operator, we assume that in the thermodynamic limit $|c(q)|$ becomes a constant in momentum space, i.e. $c(q) = \mathrm{e}^{i\alpha(q)}/\sqrt{N}$ is just a phase factor normalized by the number $N$ of spinon states. Using this in (65), one obtains for the profile

$$\mathcal{M}_j(t) = \frac{1}{N} \sum_p \sum_q \mathrm{e}^{-it(\varepsilon_s(q)-\varepsilon_s(p))} \mathrm{e}^{i(\alpha(q)-\alpha(p))} \frac{\langle p|\, \sigma_j^z\, |q\rangle}{\langle \psi_\uparrow|\, \sigma_j^z\, |\psi_\uparrow\rangle}\,. \tag{66}$$

Clearly, the main difficulty of calculating (66) is due to the form factors $\langle p|\, \sigma_j^z\, |q\rangle$. For the transverse Ising and XY chains, such form factors are known explicitly [53, 54] and were used to obtain a double integral representation of the magnetization profile [26, 28]. The hydrodynamic limit can then be obtained from the stationary-phase analysis of the integrals. Moreover, there exists a number of form factor results for the XXZ chain as well (see e.g. [55, 56]), which were used in numerical studies of the magnetization profile after a spin-flip excitation [57]. Unfortunately, however, the expressions are typically rather involved or not in a representation that could be useful for our purposes. In fact, we are not aware of any results where the required single-spinon matrix elements are evaluated as a function of the spinon rapidity or momentum.

Nevertheless, based on the known results, we give a handwaving argument about how the main structure of the form factor should look like. Most importantly, we assume that it becomes singular for $p \to q$ and can be written as

$$\lim_{p\to q} \frac{\langle p|\, \sigma_j^z\, |q\rangle}{\langle \psi_\uparrow|\, \sigma_j^z\, |\psi_\uparrow\rangle} \simeq \frac{i}{N} \mathrm{e}^{i(q-p)j} \frac{\mathcal{F}(q)}{p-q}\,. \tag{67}$$

Here the only $j$-dependence is in the exponential factor that follows from the action of the translation operator, and the function $\mathcal{F}(q)$ denotes the slowly varying part of the form factor around its pole. The factor $1/N$ is required for a proper thermodynamic limit of (66). Converting the sums into integrals, one can proceed with the stationary phase analysis similarly to the XY case [28], by expanding the phases around $Q = q - p = 0$. Using a resolution of the pole and the definition of the step function

$$\frac{1}{Q} = i\pi\delta(Q) + \lim_{\epsilon\to 0} \frac{1}{Q+i\epsilon}, \qquad \Theta(x) = -\lim_{\varepsilon\to 0} \int_{-\infty}^{\infty} \frac{\mathrm{d}Q}{2\pi i} \frac{\mathrm{e}^{-iQx}}{Q+i\varepsilon}, \tag{68}$$

one arrives at the following simple expression for the profile

$$\mathcal{M}_j(t) = 1 - 2\tilde{\mathcal{N}}, \qquad \tilde{\mathcal{N}} = \int_0^\pi \frac{\mathrm{d}q}{\pi} \Theta(v_s(q)t - j)\,\mathcal{F}(q)\,. \tag{69}$$

Note that the proper normalization of the profile for $t = 0$ requires to have

$$\int_0^\pi \frac{\mathrm{d}q}{\pi}\,\mathcal{F}(q) = 1\,. \tag{70}$$

The result (69) is nothing else but the quasiparticle interpretation of the magnetization profile in the hydrodynamic limit. Indeed, the initial sharp domain wall is carried away by spinons of different momenta $q$ and velocities $v_s(q)$, where $\mathcal{F}(q)$ gives the corresponding weight. Unfortunately, without an explicit analytical result on the form factor, one has to make a guess on the weight function. The simplest assumption is $\mathcal{F}(q) \equiv 1$, which indeed holds true for the XY chain form factors [28]. With this simple choice one actually has $\tilde{\mathcal{N}} = \mathcal{N}$, that is we recover the spinon fraction introduced in (16) for the description of the entropy profile. In Fig. 9 we show the comparison of this simple ansatz to the tDMRG data, with a rather good agreement for a large $\Delta = 5$. For $\Delta = 2$, however, one can already see the deviations from our simple ansatz, which fails completely for even smaller anisotropies. Thus, in sharp contrast to the case of the entanglement entropies, the spinon contributions to the magnetization cannot be taken to be equal, except for close to the Ising limit.

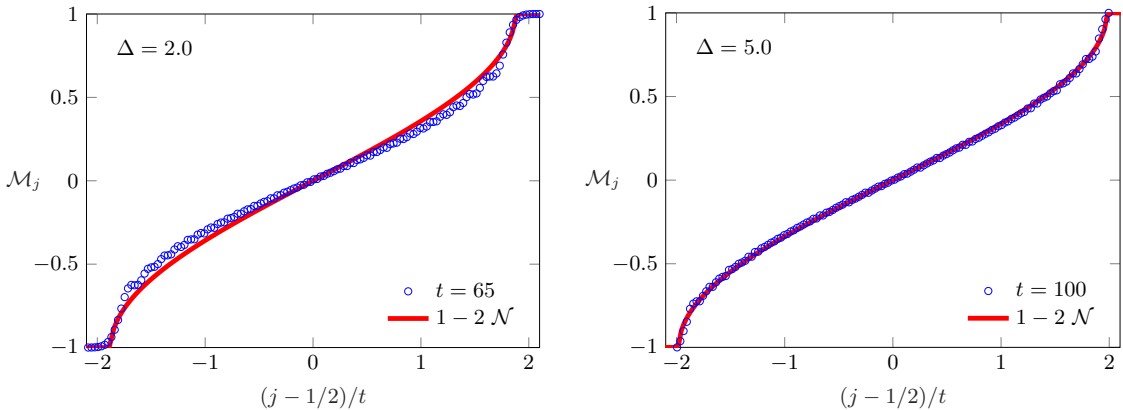

Figure 9: Normalized magnetization profiles $\mathcal{M}_j(t)$ obtained from tDMRG calculations for $\Delta = 2.0$ (left) and $\Delta = 5.0$ (right) after a domain-wall excitation $\tilde{m}_1$ in a chain of length $L = 400$. The solid lines show the ansatz $1 - 2\mathcal{N}$, with the spinon fraction Eq. (16) calculated from the velocities in Eq. (13).

# 6 Summary and discussion

We studied the entanglement spreading in the XXZ chain after excitations that are strictly local in terms of the fermion operators. In the gapless phase we found that the time evolution after a fermion creation operator yields an entropy profile that can be well described by a probabilistic quasiparticle ansatz for not too large $\Delta$, assuming equal contributions from low-lying spinon excitations. On the other hand, for a local Majorana excitation we observe that the quasiparticle ansatz holds only up to a multiplicative factor, determined by the excess entropy at the operator insertion point. This is in agreement with our CFT

calculations, which suggest that the excess entropy exceeds $\ln(2)$ for any $\Delta \neq 0$, with a very slow convergence towards the asymptotic value $2\ln(2)$. In the symmetry-broken gapped phase we considered a different Majorana excitation, creating an antiferromagnetic domain wall. For the entropy profile we find again a nontrivial prefactor, whereas our simple ansatz for the magnetization, assuming equal contributions from the spinons, holds only in the Ising limit $\Delta \to \infty$.

The main limitation of our quasiparticle ansatz (17) is that it includes only the low-lying spinons. It is natural to ask how well such an assumption actually holds for our local excitations in the different regimes. A simple way to quantify the spectral weight of the spinons in the gapless regime is via the energy difference $\langle \Delta E \rangle = \langle \psi_0 | (m_1 H m_1 - H) | \psi_0 \rangle$ of the Majorana excitation (equal to that of $c_1^\dagger$ by particle-hole symmetry) measured from the ground state, whereas in the gapped case we replace $m_1 \to \tilde{m}_1$. Our assumption in both regimes was that one can practically work with single-spinon states, whose energies above the ground state are given by the corresponding dispersions $\varepsilon_s(q)$ in (4) and (10), respectively. This yields the simple formula for the energy difference

$$\langle \Delta E \rangle = \int_0^\pi \varepsilon_s(q) \frac{\mathrm{d}q}{\pi} . \tag{71}$$

To test the validity of our assumption, in Fig. 10 we compare the energy difference obtained from DMRG to the formula (71) in both gapless and gapped phases. As expected, the result at the free-fermion point $\Delta = 0$ is exactly reproduced, while the error remains relatively small in the regime $|\Delta| \lesssim 0.5$. However, not surprisingly, the overall behaviour of $\langle \Delta E \rangle$ is not properly captured by the naive ansatz (71), especially for $\Delta \to -1$, which is exactly what we observed for the entropy profiles in Fig. 1. On the other hand, in the gapped phase shown on the right of Fig. 10, one has a qualitatively good description in the entire regime, with the error decreasing for $\Delta \gg 1$. This explains why we had a much better overall description of the entropy profiles for $\Delta > 1$ via the quasiparticle ansatz (62).

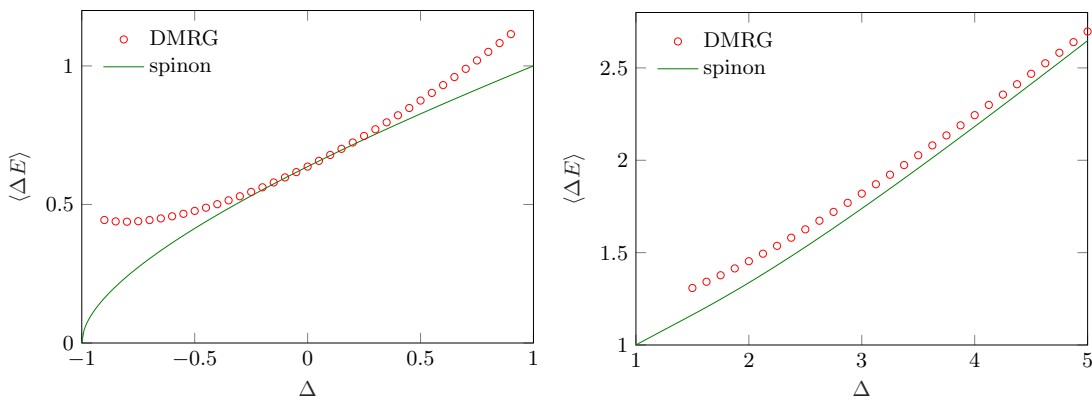

Figure 10: Energy difference due to the insertion of local operator $m_1$ in the gapless (left) and $\tilde{m}_1$ in the gapped (right) regime. DMRG results (symbols) for $L = 400$ are compared to the spinon ansatz (lines) in Eq. (71). Note the different vertical scales.

Another feature that is not completely understood is the multiplicative factor of the spinon ansatz appearing for Majorana excitations. In the gapless phase this could be accounted for the mixing of the chiral boson modes and yields a factor 2 in the limit $t \to \infty$ for any $\Delta \neq 0$. The exceptional behaviour of the XX chain can actually be also understood directly, using a duality transformation [58–61] that relates it to two independent and critical transverse Ising chains. Furthermore, as shown in [26], the Majorana excitation on

the XX chain transforms under the dual map into a Majorana excitation acting only on a *single* Ising chain. Hence, the asymptotic excess entropy is given by $\ln(2)$ and there is no doubling in this case. On the other hand, in the gapped phase the prefactor in (62) seems to be nontrivially related to the ground-state entanglement entropy. Note that a similar observation was reported after a local quench in the non-critical transverse Ising chain [62], where the entanglement plateau was also found to be related to the ground-state value. A deeper understanding of these effects requires further studies.

Finally, let us comment about the case where the locality of the excitation is not imposed in the fermionic but rather in the spin picture. In other words, instead of the $c_j^\dagger$ excitation one could consider the spin operator $\sigma_j^+$ by dropping the Jordan-Wigner string in (19). According to our tDMRG calculations carried out for this case, the entropy profiles change completely, becoming more flat in the center with a maximum that stays way below $\ln(2)$. In short, the fermionic nature of the local excitations turns out to be essential for the applicability of the quasiparticle description.

# Acknowledgements

The authors acknowledge funding from the Austrian Science Fund (FWF) through project No. P30616-N36.

# A   Correlation functions of vertex operators

In the following we give the main steps of the calculation of the excess Rényi entropy $\Delta S_2$, obtained via the ratio (34) of four-point and two-point functions. As in the main text, we consider two different local operators, the one corresponding to the fermion creation

$$\mathcal{O}_f = \mathrm{e}^{ik_F d}\psi^\dagger + \mathrm{e}^{-ik_F d}\bar{\psi}^\dagger\,, \tag{72}$$

as well as the Hermitian Majorana excitation

$$\mathcal{O}_m = \mathrm{e}^{ik_F d}\psi^\dagger + \mathrm{e}^{-ik_F d}\bar{\psi}^\dagger + \mathrm{e}^{-ik_F d}\psi + \mathrm{e}^{ik_F d}\bar{\psi}\,. \tag{73}$$

They are composed of chiral fermion fields which, after the Bogoliubov transformation (41), can be written as vertex operators (42) involving chiral boson fields. The holomorphic and anti-holomorhic components of the vertex operators are summarized in the table below, where $c = \cosh(\xi)$ and $s = \sinh(\xi)$.

|          | $\psi$ | $\psi^\dagger$ | $\bar{\psi}$ | $\bar{\psi}^\dagger$ |
|----------|--------|----------------|--------------|----------------------|
| $\alpha$ | $-c$   | $c$            | $s$          | $-s$                 |
| $\beta$  | $-s$   | $s$            | $c$          | $-c$                 |

Table 1: Parameters of the vertex operators (42) for the fermionic fields

We start by evaluating the two point function in the denominator of (34). Using the fact that vertex operators are primaries with conformal dimensions $h = \alpha^2/2$ and $\bar{h} = \beta^2/2$, one immediately obtains the nonvanishing two-point functions on the plane as

$$\langle \psi(w_1, \bar{w}_1)\psi^\dagger(w_2, \bar{w}_2)\rangle \propto (w_1 - w_2)^{-c^2}\,(\bar{w}_1 - \bar{w}_2)^{-s^2}\,,$$
$$\langle \bar{\psi}(w_1, \bar{w}_1)\bar{\psi}^\dagger(w_2, \bar{w}_2)\rangle \propto (w_1 - w_2)^{-s^2}\,(\bar{w}_1 - \bar{w}_2)^{-c^2}\,. \tag{74}$$

654   From (32) we have $w_1 - w_2 = \bar{w}_1 - \bar{w}_2 = 2\epsilon$, thus we obtain for the two-point functions

$$\langle \mathcal{O}_f^\dagger(w_1, \bar{w}_1) \mathcal{O}_f(w_2, \bar{w}_2) \rangle = 2 \, (2\epsilon)^{-(c^2+s^2)}, \qquad \langle \mathcal{O}_m^\dagger(w_1, \bar{w}_1) \mathcal{O}_m(w_2, \bar{w}_2) \rangle = 4 \, (2\epsilon)^{-(c^2+s^2)}. \tag{75}$$

655   Let us now move to the four-point function on the Riemann surface $\Sigma_2$. This is a
656   sum of many terms, from which the nonvanishing contributions allowed by the neutrality
657   conditions (44) are given by

$$\langle \psi\psi^\dagger \bar{\psi}\bar{\psi}^\dagger \rangle, \quad \langle \bar{\psi}\bar{\psi}^\dagger \psi\psi^\dagger \rangle, \quad \langle \bar{\psi}\psi^\dagger \psi\bar{\psi}^\dagger \rangle, \quad \langle \psi\bar{\psi}^\dagger \bar{\psi}\psi^\dagger \rangle, \quad \langle \psi\psi^\dagger \psi\psi^\dagger \rangle, \quad \langle \bar{\psi}\bar{\psi}^\dagger \bar{\psi}\bar{\psi}^\dagger \rangle. \tag{76}$$

658   We first analyze the Jacobian of the transformation (38) from $\Sigma_2 \to \Sigma_1$. The derivatives
659   of the mapping are given by

$$\frac{\mathrm{d}w}{\mathrm{d}z} = i\ell \frac{n z^{n-1}}{(1-z^n)^2}, \qquad \frac{\mathrm{d}\bar{w}}{\mathrm{d}\bar{z}} = -i\ell \frac{n \bar{z}^{n-1}}{(1-\bar{z}^n)^2}. \tag{77}$$

660   Introducing the variable

$$\chi = \frac{(1-z_1^2)^2 (1-z_2^2)^2}{4 z_1 z_2}, \tag{78}$$

661   one obtains for the first four contributions in (76)

$$\ell^{-2(c^2+s^2)} \chi^{c^2/2} \bar{\chi}^{s^2/2} \chi^{s^2/2} \bar{\chi}^{c^2/2} = \ell^{-2(c^2+s^2)} |\chi|^{c^2+s^2}, \tag{79}$$

662   whereas for the last two contributions we have, respectively

$$\ell^{-2(c^2+s^2)} \chi^{c^2} \bar{\chi}^{s^2}, \qquad \ell^{-2(c^2+s^2)} \chi^{s^2} \bar{\chi}^{c^2}. \tag{80}$$

663   Note that there is an extra sign factor $(-i)^{c^2}(i)^{s^2}(i)^{s^2}(-i)^{c^2} = (-i)^{2(c^2-s^2)} = -1$ which
664   multiplies the first two Jacobian.
665   The next step is to evaluate the vertex four-point functions. Using (43) this reads for
666   the first term in (76)

$$z_{12}^{-c^2} z_{34}^{-s^2} z_{13}^{-cs} z_{24}^{-cs} z_{14}^{cs} z_{23}^{cs} \bar{z}_{12}^{-s^2} \bar{z}_{34}^{-c^2} \bar{z}_{13}^{-cs} \bar{z}_{24}^{-cs} \bar{z}_{14}^{cs} \bar{z}_{23}^{cs} = (-1)|1-\eta|^{2cs} |\eta|^{-(c^2+s^2)} |z_{13} z_{24}|^{-(c^2+s^2)} \tag{81}$$

667   Note that we have used the property $z_{34} = -z_{12}$. It is easy to check that one obtains the
668   very same factor from the second term. Similarly, using $z_{23} = z_{14}$, one can check that the
669   third and fourth terms deliver

$$z_{12}^{cs} z_{34}^{cs} z_{13}^{-cs} z_{24}^{-cs} z_{14}^{-c^2} z_{23}^{-s^2} \bar{z}_{12}^{cs} \bar{z}_{34}^{cs} \bar{z}_{13}^{-cs} \bar{z}_{24}^{-cs} \bar{z}_{14}^{-s^2} \bar{z}_{23}^{-c^2} = |\eta|^{2cs} |1-\eta|^{-(c^2+s^2)} |z_{13} z_{24}|^{-(c^2+s^2)}. \tag{82}$$

670   For the fifth term one has

$$[\eta(1-\eta)]^{-c^2} (z_{13} z_{24})^{-c^2} [\bar{\eta}(1-\bar{\eta})]^{-s^2} (\bar{z}_{13} \bar{z}_{24})^{-s^2}, \tag{83}$$

671   and the last term follows by interchanging $c$ and $s$ above.
672   In order to obtain an expression in terms of the cross-ratios, one can rewrite (78) as

$$\chi = \left( \frac{\ell}{2\epsilon} \right)^2 \eta(1-\eta) \, z_{13} z_{24}. \tag{84}$$

673   Putting everything together, one arrives at the four-point function

$$2 \, (2\epsilon)^{-2(c^2+s^2)} \left[ |\eta|^{(c+s)^2} + |1-\eta|^{(c+s)^2} + 1 \right]. \tag{85}$$

674 Evaluating the four-point function for the Majorana excitation (73) is more cumber-
675 some, since one has a large number of terms to consider. There are, however, some simple
676 rules and symmetry arguments which make the task easier. First of all, one should clearly
677 always have the same number of creation and annihilation operators, for the neutrality
678 conditions (44) of the vertex correlation functions to be satisfied. This already drasti-
679 cally reduces the number of terms to consider. The remaining ones can be collected into
680 families, some of them given by (76).

681 Let us consider the family generated by the first term in (76), by allowing permutations
682 of the left- and right-moving operators separately (i.e. interchanging the first or last two
683 operators). If only the first or last two are interchanged, the vertex correlator (81) is
684 modified by replacing

$$|1 - \eta|^{2cs} \to |1 - \eta|^{-2cs}, \tag{86}$$

685 whereas the correlator remains the same if both of them are interchanged. The next
686 family is generated by the second term in (76), which is actually related to the first one by
687 Hermitian conjugation. Hence this just gives a factor of two. The same argument holds
688 for the next two families, where interchanging only one pair modifies the correlator in (82)
689 as

$$|\eta|^{2cs} \to |\eta|^{-2cs}. \tag{87}$$

690 Finally, the single interchange in the fifth family leads to

$$(1 - \eta)^{-c^2} \to (1 - \eta)^{c^2}, \qquad (1 - \bar{\eta})^{-s^2} \to (1 - \bar{\eta})^{s^2}, \tag{88}$$

691 whereas the last family follows by interchanging $c$ and $s$ above.

692 There are, however, two additional families appearing where the left- and right-moving
693 particles are intertwined. They are given by the representative correlators

$$\langle \psi \bar{\psi}^\dagger \psi^\dagger \bar{\psi} \rangle, \quad \langle \bar{\psi} \psi^\dagger \bar{\psi}^\dagger \psi \rangle. \tag{89}$$

694 Defining the variable

$$\sigma = \frac{(1 - z_1^2)^2 (1 - \bar{z}_2^2)^2}{4 z_1 \bar{z}_2}, \tag{90}$$

695 the corresponding Jacobians contain the factors $\sigma^{c^2} \bar{\sigma}^{s^2}$ and $\sigma^{s^2} \bar{\sigma}^{c^2}$, respectively. Further-
696 more, the vertex correlation functions yield

$$|\eta|^{\pm 2cs} |1 - \eta|^{\mp 2cs} (z_{13} \bar{z}_{24})^{-c^2} (\bar{z}_{13} z_{24})^{-s^2}, \qquad |\eta|^{\pm 2cs} |1 - \eta|^{\mp 2cs} (z_{13} \bar{z}_{24})^{-s^2} (\bar{z}_{13} z_{24})^{-c^2}, \tag{91}$$

697 and each term comes with a double multiplicity. Collecting all the terms, the four-point
698 function takes the form

$$2 (2\epsilon)^{-2(c^2 + s^2)} (2A + B + C), \tag{92}$$

699 where the factors $A$, $B$ and $C$ are reported in (51)-(53).

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
