# Peer review of "Entanglement spreading after local fermionic excitations in the XXZ chain"

_SciPost Physics_

## Round 1 · Referee Report · Wu-zhong Guo (Referee 1) · 2020-12-2

Strengths

  1. Local excitation for interacting model is interesting
  2. The conclusions are supported by both numerical and analytical calculations

Weaknesses

  1. Some results are vague
  2. The physical explanation is not so clear

Report

This paper investigates the dynamical evolution of local excitation of the XXZ chain model. The authors discuss the evolution of entanglement entropy for different cases including a local fermionic and Majorana excitation. The authors also consider the effect of the strength of interaction. The previous studies on this topic mainly focus on non-interacting models and some simple CFT models. The paper achieves some interesting results by numerical and analytical calculations , thus gives some important insight on the quasiparticle ansatz of local excitation. I think this paper meets the criteria of acceptance of SciPost. But I have some questions on their results: 1. The results in section 3.2. By tDMRG simulations the authors get some figures (Fig.1) on the variance of $\Delta S$ and $\xi$. They find there exists a peak around $r=0$. This seems very strange. 1). Some points are isolated from others. 2). In section 3.3 the Majorana excitation has no such peaks. I think the authors should explain why the fermionic and Majorana ones are so different. The Majorana is linear combination of two fermionic operators. For me it is more natural that they both show the peaks if the peaks are physical. More, for small strength $\Delta=0.2$ the peaks also appear. But for $\Delta=0$ (free-fermion point) is there a peak by numerical method? In this case the quasiparticle ansatz works well I think there are no peaks.
2. For $|\Delta|\le 1$ (gapless phase) the authors find the maximal $\Delta S$ are associated with coupling $\Delta$ and can be larger than $ln 2$ by numerical calculation shown in Fig.1. But in section 4.1 they find the maximal $\Delta S_2$ is independent with $\Delta $ given by $ln 2$. The authors should explain the tension between the two results. 3. About the results in Fig.5 . Could the authors explain more why the curves oscillate ? Is it from the finite size effect? 4. In the gapless phase the maximal entropy $ln 2$ of the fermionic excitation has a good explanation as EPR-like state. The Majorana excitation is linear combination of two fermionic excitation. It seems the state is given by a linear combination of two EPR-like state thus give the factor 2. If possible the authors could give more physical picture on this results.

Requested changes

see the reports.

  • validity: good
  • significance: ok
  • originality: good
  • clarity: good
  • formatting: good
  • grammar: excellent

Author:  Viktor Eisler  on 2020-12-21  [id 1096]

(in reply to Report 1 by Wu-zhong Guo on 2020-12-02)

We thank the referee for the critical reading of the manuscript, for insightful remarks and valuable suggestions. Below we try to address all the issues raised by the referee, indicating the corresponding changes to the manuscript.

1. Regarding the peaks in Fig. 1, the referee is completely right about the discrepancy. In fact, we made an error in plotting the results and became aware of it only after submission. Indeed, when considering the excess entropy $\Delta S=S(t)-S(0)$, we subtracted the value $S(0)$ after applying the excitation. In case of Majorana excitations this does not make any difference at all, since those are products of single-site unitary operators, and thus do not change the entanglement for arbitrary cuts along the chain. However, the fermion creation operator is clearly non-unitary, and its effect is to create a small dip in the entropy profile close to the excitation. Thus, the observed peaks in Fig. 1 were an artifact of subtracting this profile instead of the ground-state value, and the dip in $S(0)$ became a peak in $\Delta S$. We have now corrected the error using the ground-state value $S(0)$, and the updated Fig. 1 does not show any peaks.

2. We thank the referee for pointing out this issue, which was not properly explained in the text. In the discussion under Fig. 1, we pointed out that the entropy could only exceed $\ln 2$, if one takes into account other families of quasiparticles that exist in the spectrum of the XXZ chain. Unfortunately, taking them into account would require the knowledge of the overlaps with the initial state, which we do not have access to. On the other hand, the CFT treatment in terms of a Luttinger liquid theory can only describe the spinon excitations. This is already clear from the fact that the velocity in (28) is given by the maximum spinon velocity. Hence, the maximum of the excess Renyi entropy cannot exceed $\ln 2$, in perfect agreement with the ansatz (17) for the excess von Neumann entropy. However, calculating $\Delta S_2$ for the XXZ chain numerically, one would observe again a maximum value larger than $\ln 2$, due to contributions from other quasiparticle families. We added the following paragraph at the end of section 4.2 to clarify this issue.

"Finally, one should note that a numerical evaluation of $\Delta S_2$ in the XXZ chain gives rise again to profiles that exceed $\ln(2)$ for larger values of $|\Delta |$, similarly to what was observed for $\Delta S$ in Fig. 1. The discrepancy is due to the fact that the Luttinger liquid theory describes only the spinon modes and cannot account for other families of quasiparticles in the XXZ chain that are responsible for the enhanced excess entropy."

  1. The oscillation of $\Delta S_2$ in Fig. 5 is a simple lattice effect. In fact, similar oscillations are also present in the von Neumann excess entropy $\Delta S$ in Fig. 3, however with smaller amplitude.

4. The explanation of the difference between fermion and Majorana excitations is actually more complicated. The simple argument suggested by the referee, i.e. that the factor 2 difference is due to the linear combination of two EPR-like states is not correct. Indeed, this would only hold if the operators in the linear combination were orthogonal, which is clearly not the case here. Furthermore, in the non-interacting case $\Delta=0$ one has no factor 2, showing that this effect has no simple explanation in the fermionic picture. In fact, its origin can only be understood in the bosonic representation, where the left- and right-moving modes become mixed in the interacting case.

---

## Round 1 · Referee Report · Anonymous (Referee 2) · 2020-12-7

Report

In the work submitted for publication in SicPost Physics the authors study the time evolution of the entanglement after local fermionic excitations are created in a chain described by the XXZ model. They present numerical results obtained via density matrix renormalization group, which are initially analyzed based on a quasiparticle ansatz. To get a better description of their numerical data, they go further and present a conformal field theory based calculation. They get a general good agreement between numerical and analytical results in the regions of parameters for which both calculations apply. The manuscript is very well written: assumptions, calculations, results are all very well explained. I have few questions and comments that I detail below.

  • As in previous works, the authors consider that the spreading of entanglement is independent of the quasiparticle momentum. What is the physical justification for this assumption? What are the implications of it in the results?

  • The authors consider that the excitations are created in the middle of the chain. How do their results and conclusions depend on this specific choice? In Fig. 1, is the deviation from the quasiparticle ansatz at zeta ~ 0 related to this choice?

Minor comments:

  • In both eq. (5) and (13) the authors define the quantity v_s(q). In eq. (5), the definition applies for the gapless phase, while eq. (13) holds for the gapped phase. The authors may want to add a superscript to the quantity to differentiate the two cases.

  • In eq. (43), shouldn't V on the left side of the equation depend on the index j under the product symbol?

  • validity: -
  • significance: -
  • originality: -
  • clarity: -
  • formatting: -
  • grammar: -

Author:  Viktor Eisler  on 2020-12-21  [id 1095]

(in reply to Report 2 on 2020-12-07)

We thank the referee for the positive evaluation of our work and for the useful questions and remarks, which are addressed in detail below.

  • As in previous works, the authors consider that the spreading of entanglement is independent of the quasiparticle momentum. What is the physical justification for this assumption? What are the implications of it in the results?

The physical justification actually follows from the probabilistic interpretation. Indeed, in the equilibrium case, quasiparticle excitations are simple plane waves, implying a homogeneous probability distribution independent of the momentum. Thus the excess entanglement is given by the binary entropy function involving only the ratio of subsystem and full system lengths.

For the time evolution, the initial state should first be decomposed into the quasiparticle basis. However, considering a strictly local excitation in real space, the quasiparticle weights should be homogeneous in momentum space. Finally, the last assumption is the semi-classical propagation of the quasiparticles with their respective group velocities. Thus, at this semi-classical level, the only momentum dependence is due to the dispersion, since the initial state is an equal-weight superposition. If one had only a single type of quasiparticles, the ansatz (16) would give the probability of finding the excitation within the subsystem in this semi-classical approximation.

  • The authors consider that the excitations are created in the middle of the chain. How do their results and conclusions depend on this specific choice? In Fig. 1, is the deviation from the quasiparticle ansatz at zeta ~ 0 related to this choice?

The results are completely independent of the location of the excitation, at least if it's sufficiently far away from the chain boundaries. Indeed, the only figure that would change is the one on the right of Fig. 3, where the effect of reflections from the chain boundaries is discussed. For a non-symmetric choice, the end of the first plateau would not immediately be followed by the second one. We added the sentence "For an excitation located farther away from the chain center, the corresponding plateaus would be separated" in the last paragraph of Sec. 3 to emphasize the role of the symmetry.

Regarding Fig. 1, the deviation actually follows from an error we made in plotting the results, and became aware of it only after submission. Indeed, when considering the excess entropy $\Delta S=S(t)-S(0)$, we subtracted the value $S(0)$ after applying the excitation. In case of Majorana excitations this does not make any difference at all, since those are products of single-site unitary operators, and thus do not change the entanglement for arbitrary cuts along the chain. However, the fermion creation operator is clearly non-unitary, and its effect is to create a small dip in the entropy profile close to the excitation. Thus, the observed peaks in Fig. 1 were an artifact of subtracting this profile instead of the ground-state value, and the dip in $S(0)$ became a peak in $\Delta S$. We have now corrected the error using the ground-state value $S(0)$, and the updated Fig. 1 does not show any peaks.

  • In both eq. (5) and (13) the authors define the quantity v_s(q). In eq. (5), the definition applies for the gapless phase, while eq. (13) holds for the gapped phase. The authors may want to add a superscript to the quantity to differentiate the two cases.

We were originally thinking about using a superscript, but then decided not to, as this would just lead to a more complicated notation. Since the discussion of the gapped and gapless cases is completely separated in the text, we believe this would not lead to a confusion.

  • In eq. (43), shouldn't V on the left side of the equation depend on the index j under the product symbol?

We corrected the typo.

---

## Editorial Decision

resubmitted